# The return to 1980 stratospheric halogen levels: A moving target in ozone assessments from 2006 to 2022

Megan J. Lickley[1,2], John S. Daniel[3], Laura A. McBride[4], Ross J. Salawitch[5,6,7], Guus J.M. Velders[8,9]

[1]Earth Commons, Georgetown University, Washington, DC 20057, USA
[2]Science, Technology, and International Affairs Program, Georgetown University, Washington, DC 20057, USA
[3]NOAA Chemical Sciences Laboratory, Boulder, CO, 80305, USA
[4]Department of Chemistry and Biochemistry, Albright College, Reading, PA 19612, USA
[5]Department of Chemistry and Biochemistry, University of Maryland College Park, College Park, MD, 20740, USA
[6]Department of Atmospheric and Oceanic Science, University of Maryland College Park, College Park, MD, 20740, USA
[7]Earth System Science Interdisciplinary Center, University of Maryland College Park, College Park, MD, 20740, USA
[8]National Institute for Public Health and the Environment (RIVM), Bilthoven, the Netherlands
[9]Institute for Marine and Atmospheric Research Utrecht, Utrecht University, Utrecht, the Netherlands

*Correspondence to*: Megan J. Lickley (megan.lickley@georgetown.edu)

**Abstract.** The international scientific assessment of ozone depletion is prepared every four years to support decisions made by the Parties to the Montreal Protocol. In each assessment an outlook of ozone recovery time is provided. The year when equivalent effective stratospheric chlorine (EESC) returns to the level found in 1980 is an important metric for the recovery of the ozone layer. Over the past five assessments, the expected date for the return of EESC to the 1980 level,
for mid-latitudes, has been delayed, from year 2049 in the 2006 assessment to 2066 in the 2022 assessment, which represents a delay of 17 years over a 16-year assessment period. Here, we quantify the primary drivers that have delayed the expected EESC recovery date between each of these assessments. We find that by using identical EESC formulations the delay between the 2006 and 2022 assessments' expected return of EESC to 1980 levels is shortened to 12.6 years. Of this delay, bank calculation methods account for ~4 years, changes in the assumed atmospheric lifetime for certain ODSs
account for ~3.5 years, an under-estimate of the emission of carbon tetrachloride accounts for ~3 years, and updated historical mole fraction estimates of ODSs account for ~1 year. Since some of the underlying causes of these delays are amenable to future controls (e.g. capture of ODSs from banks and limitations on future feedstock emissions), it is important to understand the reasons for the delays in expected recovery date of stratospheric halogens.

## 1 Introduction

The Montreal Protocol is often lauded as the signature global environmental success story. Since its entry into force in 1989, it has led to large reductions in the production of ozone depleting substances (ODSs) globally and avoided a world with

substantial ozone loss (Newman et al. 2009; Morgenstern et al. 2008). To inform potential policy decisions of the Parties to the Protocol, every 4 years a Scientific Assessment of Ozone Depletion (SAOD) report is prepared by leading international

experts in atmospheric science and related fields under the auspices of the World Meteorological Organization (WMO) and the United Nations Environment Program (UNEP). A key component of each SAOD report is an outlook of the timeline for recovery of the ozone layer. In addition to calculating the return of ozone, itself, return dates to 1980 levels of equivalent effective stratospheric chlorine (EESC) are estimated and provided, given the best scientific understanding of atmospheric processes and the assumption of global compliance with the controls of the Protocol at that time. EESC is a measure of the

abundance of stratospheric inorganic chlorine and bromine and is a proxy for the chemical depletion of stratospheric ozone by halogens (discussed further in Section 2.1). While ozone depletion began before 1980, the return of EESC to the amount that had existed in 1980 is an adopted benchmark for the path to recovery. Over the past 16 years, the expected return of EESC to the 1980 level has been pushed back from 2049 in the 2006 SAOD (Daniel and Velders et al. 2007) to 2066 in the 2022 SAOD (Daniel and Reimann et al., 2022), a delay in the expected recovery of stratospheric chlorine and bromine of 17 years over a

16-year assessment period (Fig. 1a). The reasons for this expected delay in the return of stratospheric halogens to the 1980 level have not been fully elucidated and changes to the Montreal Protocol do not explain this discrepancy, as the 2007 Montreal Amendment to the Protocol was the last major amendment with appreciable effects on EESC (e.g., see Fig Q14-1 of Salawitch et al., 2018). However, the newer EESC formulation (Engel et al., 2018) first used in the 2018 SAOD may play an important role, as the return to 1980 levels was delayed by more than a decade simply from using this newer approach (WMO, 2018).


When projections of future mole fractions of ODSs and EESC recovery are updated, the underlying causes of any changes are important to understand. Changes can relate, for example, to an updated estimate of the global atmospheric lifetime of an ODS, new estimates of the amount of an ODS in banks, more extensive controls on the future production of an ODS, or the detection of the unexpected emission of an ODS. Here, the term "bank" refers to the amount of ODS stored in existing equipment,

chemical stockpiles, foams, and other products with the potential of release to the atmosphere. An example of the unexpected emission was described by Montzka et al. (2018), who reported a slowdown in the rate of decline of atmospheric CFC-11 that they attributed to new, unreported production in eastern Asia. This study brought into question the extent to which illicit production of CFC-11, inaccuracies in the assessed emission of CFC-11 from banks, variability in atmospheric transport(Ray et al. 2020), or possibly even inaccuracies in the atmospheric lifetime of CFC-11 were contributing to the delay in the expected

decline of the global mean mole fraction of CFC-11 between the projection given in the 2006 SAOD and data presented in the 2014 SAOD. Subsequent studies confirmed the Montzka et al. (2018) findings by identifying emissions of CFC-11 originating from eastern Asia, likely resulting from new production in breach of the Montreal Protocol (Rigby et al., 2019; Benish et al., 2021; Montzka et al., 2021; Park et al., 2021). Furthermore, an upward revision in the magnitude of the bank of ODSs (Lickley et al., 2020; Lickley et al., 2022) led to the recent realization that the return of EESC to the 1980 level would not occur as fast

as once expected (Daniel and Reimann et al., 2022). These studies underscore the role that SAOD reports have played in setting expectations for future mole fractions of ODSs. Considering that SAODs are assembled, in part, to provide the Parties with

our best knowledge of the effectiveness of current control measures, and also to set an expectation of the future recovery of the ozone layer, a careful analysis of the drivers changing EESC projections between these past SAODs is warranted.

Here, we quantify the reasons for the delay in the projected recovery of EESC to the 1980 level given by the five most recent SAOD reports (Daniel and Velders et al., 2007; Daniel and Velders et al., 2011; Harris and Wuebbles et al., 2014; Carpenter and Daniel et al., 2018; Daniel and Reimann et al., 2022). We henceforth refer to these five reports as the 2006; 2010; 2014; 2018; and 2022 SAODs. Specifically, we consider how the delay has been affected by the consistent underestimate of the global emissions of ODSs (Montzka et al. 2018; Rigby et al. 2019; Park et al. 2018; M. J. Lickley et al. 2022; Gamlen et al.
1986)as modeled through production, banks and feedstocks, and observed in part by measured mole fractions following the publication of each SAOD. In addition, we consider the role of changes in ODS atmospheric lifetime assumptions that affect future atmospheric abundances (SPARC, 2013), and variations in the scientific understanding of the best underlying approach used to compute EESC (Newman et al. 2007; Engel et al. 2018). We begin with the 2006 SAOD because the knowledge of the release of ODSs from banks has evolved considerably since the publication of SAOD reports prior to 2006 (IPCC/TEAP,
2005). In addition, the 2006 SAOD report is the first to distinguish the return to 1980 for EESC of both mid-latitude and polar air, even though identical fractional release factors for ODSs were used for both regions. For completeness, the return to 1980 dates for EESC provided in the past eight SAOD reports are given in Supplemental Table S1.

We quantify the contributions from each of these primary drivers that have delayed the expected return to the 1980 date of
EESC between each consecutive SAOD report from 2006 to 2022. To do so, we first re-evaluate each SAOD's historical and future EESC calculations using a common formulation for EESC (Engel et al. 2018). We are then able to compare the differences in projections of EESC due to changes in historical and projected atmospheric mole fractions alone, rather than confounding the issue with different formulations for the computation of EESC. Next, we identify the primary gases driving each change in the EESC return date between consecutive SAOD reports. We then isolate the effects of four primary gases
(CFC-11, CFC-12 and halon-1301 and carbon tetrachloride (CTC)) on EESC, as changing projections of these four compounds explain ~90% of the delay in expected EESC recovery from the 2006 SAOD to the 2022 SAOD. While the other 12 gases have led to both positive and negative changes in EESC between SAODs, their overall net contribution to the return of EESC to 1980 has been substantially smaller than the contribution of the primary four gases. Therefore, our focus is on quantifying the impact of changes in the mole fraction projections of CFC-11, CFC-12 and halon-1301 and CTC on EESC. In Section 2,
we detail the calculation of EESC given in each SAOD report, including various modeling assumptions. In Section 3, we present our methods for ODS selection and for quantifying each modeling component's contribution to delaying EESC return dates. We present the results of our analysis in Section 4. Finally, we discuss the implications for future assessments in Section 5.

**2 A review of SAOD calculations for estimating EESC return dates**

The modeling approach used in the SAODs from 2006 to 2022 estimates the return of EESC to the 1980 level as a multi-step process. First, this estimate requires knowledge of pre-1980 atmospheric mole fractions of the 16 most abundant ODSs (see Table 2 for the complete list of gases) to establish the 1980 baseline level of EESC. Next, projections of future atmospheric mole fractions require assumptions about expected emissions from future production as well as emissions from existing and future equipment (termed 'banks'), along with an estimate of the atmospheric lifetime for each gas. Further, four additional years of observations between each assessment has required updating various assumptions, such that the modelled atmospheric mole fractions of ODSs are consistent with new observations. Finally, once a historical and future time series of mole fractions has been constructed for each of the 16 primary ODSs, they are aggregated together with a calculation of EESC. Each step in this modeling process has been updated between various SAODs, reflecting the best scientific knowledge at the time of publication. A summary of the different modeling assumptions is provided in Table 1, along with the lifetimes of the four most important ODSs with regards to the variations in the return of EESC to the 1980 level across the assessments. In Table 1, the term FRF refers to fractional release factor, a quantity that (as explained below) represents the conversion from organic to inorganic chlorine of each ODS.

**Table 1:** Summary of key assumptions by Scientific Assessment of Ozone Depletion report

| | 2006 SAOD | 2010 SAOD | 2014 SAOD | 2018 SAOD | 2022 SAOD |
|---|---|---|---|---|---|
| **EESC Formulation** | FRFs: Daniel et al. (1995) Age: 3-yr delta function | FRFs: Newman et al. (2007) Age: 3-yr delta function | FRFs: Newman et al. (2007) Age: Newman (2007) spectrum | **Main text** FRFs: Newman et al. (2007) Age: Newman (2007) spectrum **Appendix**: FRFs: Engel et al. (2018); Age: Engel (2018) spectrum[1] | FRFs: Engel et al. (2018) Age: Engel (2018) spectrum |
| **Lifetimes (yrs)** | | | | | |
| CFC-11 | 45 | 45 | 52 | 52 | 52 |
| CFC-12 | 100 | 100 | 102 | 102 | 102 |
| Halon-1301 | 65 | 65 | 72 | 72 | 72 |
| CTC | 26 | 26 | 26 | 32 | 30 |
| **Bank method** | | | | | |

| | | | | | |
|---|---|---|---|---|---|
| CFC-11 | Hybrid model: 2002/2015 ref years | Hybrid model: 2008 ref year | Hybrid model: 2008 ref year | Hybrid model: 2008 ref year | Bayesian model |
| CFC-12 | 2002/2015 ref years | 2008 ref year | 2008 ref year | 2008 ref year | Bayesian model |
| Halon-1301 | 2002/2015 ref years | 2008 ref year | 2008 ref year | 2008 ref year | Bayesian model |
| CTC | Not banked | Not banked | Not banked | Not banked | Not banked |
| **Future production** | | | | | |
| CFC-11 | Global production ends in 2010 | Global production ends in 2010 | Global production ends in 2010 | Global production ends in 2010 | Unexpected production is accounted for up to 2018 |
| CFC-12 | Global production ends in 2010 | Global production ends in 2010 | Global production ends in 2010 | Global production ends in 2010 | Global production ends in 2010 |
| Halon-1301 | Global production ends in 2010 | Global production ends in 2010 | Global production ends in 2010 | Global production ends in 2010 | Global production ends in 2010 |
| **Future emissions** | | | | | |
| CTC | Linear decline from 2005 top-down derived emissions to zero from 2015 onwards | 2009-2050: 6%/year decline 2050 onwards: zero emissions | 2013-2100: 6.4%/year decline | 2017-2100: 2.5%/year decline | Linear decline from 2020 top-down emissions to 15 Gg/year in 2030. 2030 onwards: 15 Gg/year |

[1]The width of the age distribution was taken from Newman et al. (2007).


Below, we review the calculation of EESC given in each SAOD from 2006 to 2022. We then present the method used for projecting future ODS atmospheric mole fractions and review each SAOD's input parameters for these calculations. While pre-1980 mole fractions of some of the 16 primary ODSs have been modified at times between assessments, this change

represents a small fraction of total change in the EESC recovery time over the entire 16-year period, and is therefore not the focus of the present analysis.

## 2.1 EESC calculations

Equivalent effective stratospheric chlorine (EESC) is a metric that has been developed to relate the surface level atmospheric abundance of ODSs to inorganic halogen loading in the stratosphere, and thus to stratospheric ozone depletion. EESC was first introduced in Daniel et al., (1995), drawing in large part from the understanding gained in Solomon et al. (1992) and Solomon and Albritton (1992). EESC weighs surface mixing ratios of ODSs with their number of Cl (or Br) atoms and their factional release factors  It has since been refined with increasing specificity with regards to the timing of halogen releases and transport lag times, discussed further below.  Methods for estimating EESC have generally followed the functional form:

$$EESC(t) = a\left(\sum_{Cl} n_i f_i \rho_i + \alpha \sum_{Br} n_i f_i \rho_i\right), \tag{1}$$

where $n_i$ is the number of chlorine or bromine atoms of an ODS, $f_i$ represents the value of the fractional release factor (FRF) of an ODS relative to CFC-11, and $\rho_i$ is the mean stratospheric mole fraction that would be expected in the absence of chemical loss for the location of interest at time $t$. Values of $\rho_i$ can be related to the surface level mole fraction for gas, $i$, by

$$\rho_i(t) = \int_{-\infty}^{t} \rho_{i,entry}(t')G(t - t')dt' \tag{2}$$

where $\rho_{i,entry}$ is the mixing ratio of the source gas at the time of entry into the stratosphere (Newman et al., 2007). The quantity $\alpha$ is the efficiency of ozone destruction by bromine radicals relative to the efficiency by chlorine radicals, which is commonly set to 60 for EESC of mid-latitude air and 65 for EESC in polar regions (Sinnhuber et al., 2009). The quantity $a$ represents the fractional release of CFC-11 (Daniel et al., 1995; Newman et al., 2007). For the 2006 SAOD report, a single set of FRFs for the global stratosphere was used. From 2010 onward, two sets of FRFs were used: one for the mid-latitude stratosphere and another for the polar stratosphere. $G$ represents the distribution of times required to be transported from entry into the stratosphere to the region of interest and is referred to as the age spectrum.  This transport time is referred to as the age-of-air of an air parcel and represents the amount of time since the parcel was last in the troposphere(Kida 1983).

The 2006 and 2010 SAODs adopted the Daniel et al. (1995) EESC calculation approach, where $G$ was assumed to be a delta function with a 3-year lag, so $\rho_i$ represented a simple 3-year time lag from surface mole fractions (then adjusted by the factors described in the previous paragraph). The 2014 and 2018 SAODs adopted the Newman et al. (2007) formulation of EESC

projections, which modified equation (1) such that both $f_i$ and $\rho_i$ were time-weighted averages, reflecting the non-linear dependence of these terms on the age-of-air in the stratosphere. In the 2018 SAOD, the Engel et al. (2018) formulation (which employs slightly different fractional release values and a different age spectrum), is adopted in Chapter 1 on historical estimates of EESC and in an appendix of Chapter 6, applied to future projections. The 2022 SAOD adopted the Engel et al. (2018) formulation for computation of EESC for both historical and future projections. The most significant difference introduced by

the Engel et al. calculation of EESC is that it attempts to weight the age spectrum by the time when the source gas dissociates, rather than using the Newman et al. (2007) approach (and the delta function approach) in which the age spectrum is identical to the age spectrum of an inert tracer. For EESC, this change results in higher weighting of air with *longer* transit times through the stratosphere and lower weight to air with *shorter* transit times (for which ODSs have been dissociated to a lesser degree, particularly in the midlatitudes), than found using the approach of Newman et al. (2007). In summary, using the new

formulation, EESC lags the troposphere more strongly than an inert tracer would.

## 2.2 Projecting ODS mole fractions

To calculate values of $\rho_i$ in equations (1) and (2), the SAODs each began with a time series from between 1951 and 1955 to 2100 of surface mole fractions for each ODS included in Table 2. The historical range of this period is developed using

observed mole fractions, when available. In recent decades, these are highly precise and accurate atmospheric observations from the AGAGE (Prinn et al., 2018) and NOAA (https://gml.noaa.gov/dv/site/) networks. Before routine and global atmospheric observations were available, there are observations from firn samples that can help constrain prior mole fractions for some of the compounds (Laube et al. 2014; Butler et al. 1999), particularly those with strong natural sources such as methyl bromide and methyl chloride. For the remaining ODSs, historical mole fractions between 1950 and 1980 were based on model

calculations using the Alternative Fluorocarbons Environmental Acceptability Study (AFEAS, 2001) reported production data, and adjusting calculations to avoid discontinuities between modeled and observed mole fractions when observations first began for each compound. The projections of future mole fractions generally consider a range of future policy scenarios, where the baseline scenario reflects the current controls. We only consider the baseline scenarios here for comparison across SAOD reports. Baseline projections begin in the year prior to publication of the assessment, $t_0$, where an initial mole fractions

$[ODS]_{i,t_0}$, for ODS, $i$, is taken from observed surface mole fractions values. Each subsequent year, $t$, is then forward simulated using a 1-box model of the atmosphere following equation (3):

$$[ODS]_{i,t} = exp\left(-\frac{1}{\tau_i}\right) \times [ODS]_{i,t-1} + A_i \times Emiss_{i,t-1} \times \tau_i \times \left(1 - exp\left(-\frac{1}{\tau_i}\right)\right), \tag{3}$$

where $\tau_i$ is the atmospheric lifetime, and $A_i$ is a conversion factor relating emissions, $Emiss_{i,t-1}$, to surface mole fractions for

gas, $i$, assuming all of the emission is immediately deposited into the atmosphere. Values of $Emiss_{i,t}$ are modeled as the sum of emissions from expected production and banks, and are iteratively simulated using equations (4) and (5):

$$Emiss_{i,t} = RF_i \times Bank_{i,t} + DE_i \times Prod_{i,t}, \qquad (4)$$

where $RF_i$ is the yearly release fraction of the bank, $Bank_{i,t}$, and $DE_i$ is the fraction of production, $Prod_{i,t}$, for ODS, $i$, in year, $t$, that is emitted in the same year as the production. The size of the bank is then updated;

$$Bank_{i,t+1} = (1 - RF_i) \times Bank_{i,t} + (1 - DE_i) \times Prod_{i,t} \qquad (5)$$

Therefore, future ODS mole fraction projections rely on assumptions about global lifetimes, bank sizes, bank release fractions, future production, and direct emissions from production. The different methods across assessments for the values used in equations (1) – (4) are further summarized below.

## 2.3 Atmospheric Lifetime assumptions

Atmospheric lifetimes of the ODSs are an important component of the projections of future mole fractions. For each SAOD report, an "assessed" best estimate for the lifetime of each compound is presented. Here, lifetime is defined as the global atmospheric mass, or burden, of a compound divided by the loss rate integrated over the entire atmosphere (SPARC, 2013). These lifetime estimates were calculated using numerous lifetime inference methods (see the Stratosphere-troposphere Processes and their Role in Climate (SPARC, 2013) report for more details). Lifetimes have been based on satellite-derived methods which convolve stratospheric distributions (as a function of altitude and pressure) of long-lived gases with photolysis rates of their destruction (Minschwaner et al., 1993), model inversion methods using ground-based measurements with prescribed emissions (Rigby et al., 2013), or tracer-tracer methods, which relate the slope of mixing ratio of a particular species to the mixing ratio of another species with a well-established lifetime (Plumb and Ko 1992). However, for many species, modelled lifetimes alone inform the atmospheric lifetimes used in the SAODs (as reported in SPARC, 2013). For CFCs and halons, atmospheric loss occurs primarily in the stratosphere through photolysis. For gases such as methyl chloroform that undergo removal in the troposphere due to processes such as reaction with OH, the lifetime may be revised due to better knowledge of the rate constant for reaction with OH, the average OH concentration itself, as well as additional years of data from which the lifetime is inferred (Prinn et al. 2001; Montzka et al. 2011). A tabulation of the lifetime of the 16 major ODSs, from the five most recent SAOD reports, appears in Supplemental Table S3. Lifetimes most central to our analysis are repeated in Table 1.

## 2.4 Banks modeling
An ODS bank refers to the quantity of gas contained in equipment or applications that is subject to later release. One approach to estimating the size of an ODS bank in a given year requires knowledge about how much of an ODS has been cumulatively

produced and released prior to the year of interest. The difference is assumed to be residing in the bank; this approach is referred to as a top-down approach (e.g. Montzka and Fraser et al., 2003). An alternative method involves estimating the quantity of equipment and/or applications in a given year that contain a particular ODS and how much ODS resides in each piece of equipment/application; this approach is referred to as a bottom-up approach (Ashford et al., 2004, Campbell et al., 2005). Due to uncertainties in data and modeling assumptions, each method yields bank estimates with significant uncertainties. The various bank estimates for CFC-11, CFC-12, and halon-1301 from the 2006 to 2022 SAODs are shown in Fig. 1.

In earlier assessments (e.g. the 2002 SAOD; Montzka and Fraser et al. 2003), projections of EESC were based on banks found using a top-down approach, where banks were estimated as;

$$Bank_{i,t+1} = Bank_{i,t} + Prod_{i,t} - Emiss_{i,t} \tag{6}$$

For the 2002 report, the production values, $Prod_{i,t}$, came from AFEAS (2001) and UNEP's Ozone Secretariat, and emissions, $Emiss_{i,t}$, were derived from observed mole fractions by rearranging equation (3). Banks were calculated by starting in the first year of production, with $Bank_{i,t_0}$ equal to zero and iterating forward.

Due to the inherent large uncertainties associated with this approach, since the bank is a small difference between two large numbers (cumulative production and cumulative emission), and due to the large discrepancies between bank estimates using this top-down approach and other bottom-up accounting methods (Daniel et al., 2007), the 2006 SAOD used a hybrid modeling approach to estimate banks. Bank estimates for 2002 and 2015 were adopted from the Technology and Economic Assessment Panel bank estimate (IPCC/TEAP, 2006), which estimated banks using a careful bottom-up calculation of inventory and expected release rates by application type. Such bottom-up estimates were not previously available. Equation (5) was then used to solve for $RF_i$, assuming $DE_i$ equal to $RF_i$, such that the 2002 and 2015 banks matched the prescribed 2002 and 2015 values, while also accounting for the reported production during this period. After solving for $RF_i$ values, banks and emissions were simulated from 2015 onwards using equations (4) and (5) and assuming $RF_i$ remained constant into the future.

This method was modified in the 2010, 2014 and 2018 SAODs, which started with a bottom-up estimate for the 2008 bank from a TEAP 2009 report (Kuijpers & Verdonik, 2009). Using equation (6), banks were calculated beginning in 2008 (forward and backward in time if necessary) for the 7 most recent years in which mole fraction observations were available. Emissions for equation (6) were calculated using equation (3), rearranged to solve for emissions. $RF_i$ was then estimated for each of these years by setting $DE_i$ equal to $RF_i$ in equation (4) and solving for $RF_i$. For the 2014 and 2018 SAODs, the average $RF_i$ value

over these 7 years was then used to project banks and emissions from 2008 onwards using equations (3) and (4). For the 2010 SAOD, a $RF_i$ that was consistent with the values over the previous 5-10 years, depending on compound, was used.

The 2022 SAOD bank estimates followed a Bayesian bank estimation method from Lickley et al. (2020, 2022). This method
develops prior distributions of the input parameters to equation (4) to account for uncertainties in production, as well as uncertainties in $RF_i$ and $DE_i$ values. Equations (3 - 5) are then simulated starting in the first year of production for the respective ODS to the end of the observational record, which results in a joint prior distribution of banks, emissions, and atmospheric mole fractions. Joint posterior distributions are obtained by updating the prior with available global averaged observed atmospheric mole fractions. This approach resulted in larger bank estimates than used in previous assessments,
largely due to the allowed possibility that ODS production was higher than reported; this higher posterior production primarily occurs because plausible RF values along with reported production values were inconsistent with atmospheric mole fractions, given the model assumptions. Projected ODS mole fractions were then estimated using posterior bank and $RF_i$ estimates in 2021 and forward simulating equations (4) and (5), with an assumed production timeseries in line with the controls set by the Montreal Protocol.

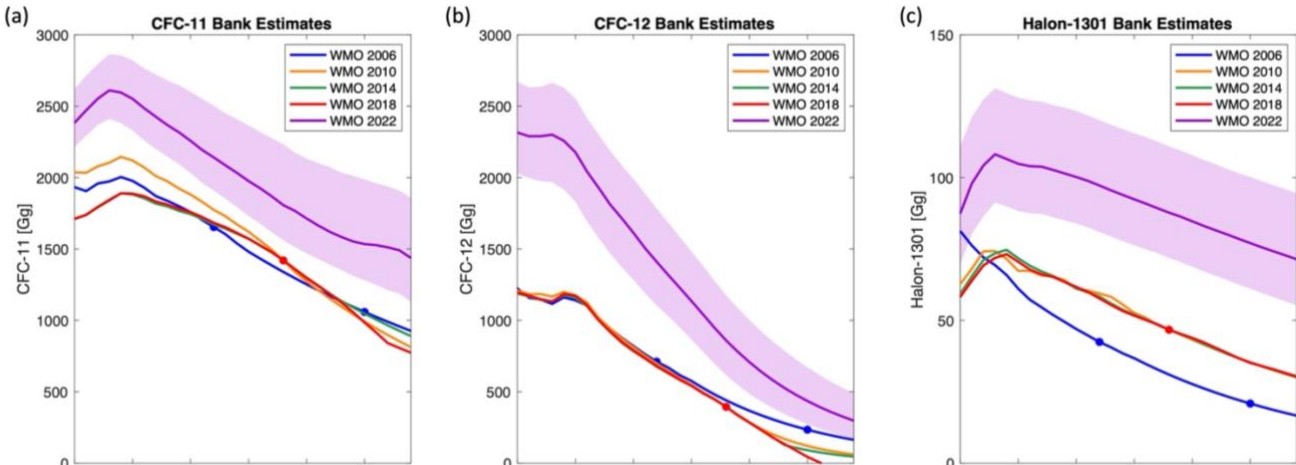

**Figure 1.** Bank estimates (a) CFC-11 (b) CFC-12 and (c) halon 1301 for each scientific assessment of ozone depletion from 2006 to 2022, as indicated by the WMO date in the legend. The blue does represent the bottom-up derived banks from IPCC/TEAP (2006) that are adopted as an initial starting point for the WMO (2006) bank assessment. The red dot represents
the starting point for the WMO (2010, 2014, 2018) assessments, taken from the 2009 TEAP report (Kuijpers & Verdonik, 2009). The shaded region for WMO (2022) represents the 5th and 95th percentile confidence bounds around the median bank estimate.

## 2.5 Carbon tetrachloride modeling

We consider CTC separately from the banked ODSs as it is not thought to be a substantially banked chemical and global emissions have been much less well understood. Therefore, CTC projections have been developed using an independent method compared to banked ODSs. The CTC budget remains an area of substantial uncertainty (SPARC, 2016), as noted in each SAOD report from 2010 to present. Under the Montreal Protocol, CTC was scheduled to be phased out of production for dispersive uses by 2010, which was consistent with near-zero country-reported production values from that time onward. Assuming CTC global production would follow the scheduled phaseout, the 2006 SAOD report adopted a future emissions pathway that began with a linear decrease from top-down derived emissions values in 2005 of 65 Gg/year to zero in 2015 and beyond. However, the expected rate of decline was not observed to be as quick as projected in the 2006 SAOD, which led to adjustments in CTC projections in subsequent SAODs. The emission estimates for CTC used in the 2010 SAOD report and the subsequent three reports are shown in Fig. 2.

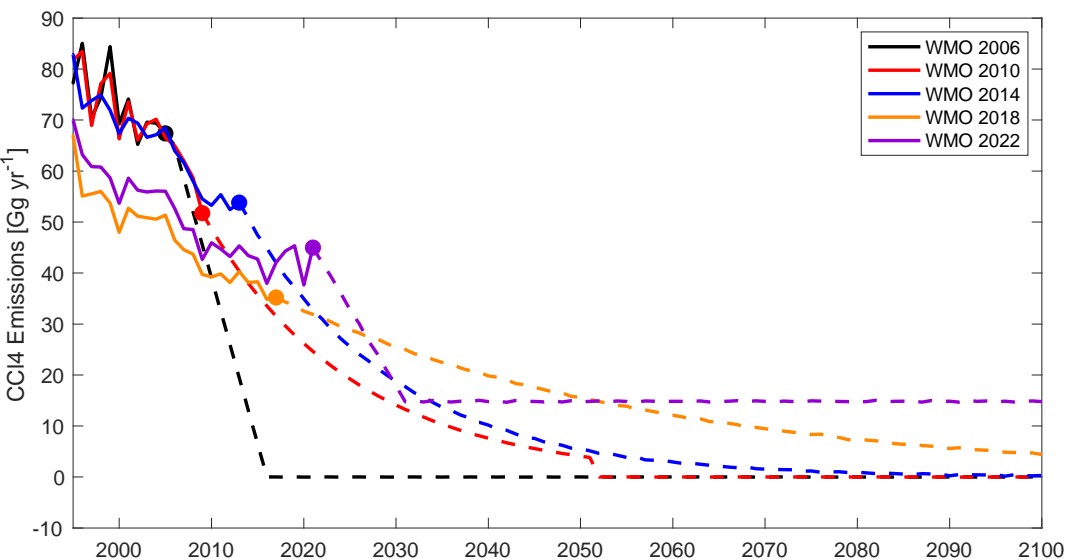

**Figure 2.** Carbon tetrachloride emissions estimates from each scientific assessment of ozone depletion (SAOD) from 2006 to 2022, as indicated by the WMO date in the legend. Solid lines represent observationally-derived emissions using the assumed lifetime from the corresponding SAOD report. The dashed line represents the emissions projection estimates from each SAOD and the dots indicate the year of each publication, which separates the observationally-derived from projected emissions.

The 2010 and 2014 SAOD reports developed CTC emissions projections by extrapolating the top-down derived CTC emissions trend from the previous five years, equivalent to 6%/year and 6.4%/year decrease in emissions, respectively. The 2010 SAOD assumed zero emissions following 2050, whereas the 2014 SAOD assumed the continued 6.4%/year decrease in emissions

(Figure 2). However, observationally-derived emissions values from 2000 – 2012, after accounting for updated atmospheric lifetime estimates, again did not reflect this assumed decline in emissions but rather were estimated to be relatively stable at $39^{45}_{34}$ Gg/year (Liang et al. 2014), highlighting the gap in understanding of emissions sources.

This projection assumption for CTC was again updated in the 2018 SAOD to better match top-down derived emissions, which were declining slower than expected at a rate of 2.5%/year over the previous two decades. This slower than expected decline was only partially explained from adjusted estimates in atmospheric lifetimes between the 2014 and 2018 SAODs (Table 1). Additional discrepancies were documented in the SPARC (2016) special report on the "mystery" of CTC, which pointed to previously unaccounted for by-product emissions during chloromethane and perchlorethylene (PCE) production, feedstock emissions for hydrofluocarbons (HFCs) and PCE, where feedstocks refer to chemicals used in the process of manufacturing different chemicals, legacy emissions including from contaminated soils and landfills, and inadvertent emissions (Sherry et al., 2018, SPARC, 2016). Sherry et al. (2018) estimated bottom-up emissions in 2014 that included ~15 Gg/year from unreported non-feedstock and fugitive emissions and ~10 Gg/year of legacy emissions from chloro-alkali plants.

The 2022 SAOD thus adopted a hybrid CTC emissions projection that began with top-down derived emissions equal to 45 Gg/year in 2020 and assumed a linear decrease in emissions from 2020 to 2030, with constant emissions of 15 Gg/year from 2030 onwards. The emissions pathway reflects an assumption that legacy emissions will decline linearly until their cessation in 2030, and an assumption of continued constant emission from, for example, feedstock sources. New knowledge of pathways of atmospheric emission of CTC continue to emerge (Li et al., 2024), which will likely result in further adjustments to future emissions in subsequent SAOD reports.

## 3 Quantifying drivers of delayed EESC return dates

Here we explain the various steps that underlie our process for quantifying the contribution of updated modeling assumptions in delaying the return of EESC to the value found for January, 1980. We use the beginning of 1980 as a marker for the return of EESC, since the return of EESC and atmospheric levels of ozone to the "pre-1980 value" is a commonly adopted metric for assessing the path to recovery of the ozone layer in the SAOD reports.

*Step 1: Update EESC calculation method to that used in WMO (2022)*

We recalculate the EESC time series for each assessment by applying the Engel et al. (2018) formulation, using each SAOD's original time series of atmospheric mole fractions for the 16 major ODSs given in Table 1. This calculation is performed first so changes in the EESC computation method do not confound the interpretation of changes in ODS projections.

*Step 2: Identify the primary gases delaying EESC return dates*

We then identify the primary ODSs that have driven the delay in EESC return dates. This identification is done by beginning with the original assessment's time series for all ODSs. We substitute in the subsequent assessment's time series and re-calculate EESC and its expected 1980 return date one gas at a time with replacement. The ODSs that dominate the delay in EESC and are also subject to banking (that is, CFC-11, CFC-12, halon-1301) are then considered in Steps 3 – 5. CTC, which is the most critical ODS in delaying expected EESC return dates between SAODs and is not assumed to be significantly

banked, is then considered in Step 6.

*Step 3: Update Lifetime assumptions to 2022 SAOD for most important banked gases*

For CFC-11, CFC-12 and halon 1301, we adopt the 2022 SAOD atmospheric lifetimes, and recalculate emissions and banks following each assessment's original bank calculation method described earlier. We then use the updated bank and release

fractions, along with the new atmospheric lifetimes to project mole fractions to 2100. See Supplementary Table S3 for a summary of the atmospheric lifetimes adopted for each SAOD.

*Step 4: Update mole fraction observations to WMO (2022) for most important banked gases*

We recalculate atmospheric mole fractions by updating the bank and emissions calculations based on observed mole fractions

out to 2021, while retaining the approaches of the respective SAODs. This step is done to evaluate the extent to which differences in the projected mole fractions from 2006-2021 would have impacted the EESC return date using the original bank and emissions estimation methods.

*Step 5: Update banks to WMO (2022) for most important banked gases*

After updating mole fractions and lifetimes, bank emissions are the only remaining discrepancy for CFC-11, CFC-12 and halon-1301. Therefore, we next update the entire projection time series using the 2022 SAOD 2020 bank values to account for the outstanding update, which is the updated bank values and approach. This step allows us to quantify the new estimated bank contributions to differences in the expected return of EESC to the 1980 level for these three gases.

*Step 6: Update CTC lifetimes and emissions projections*

CTC is treated separately from CFC-11, CFC-12 and halon 1301 because CTC is not a banked ODS; the sources of its emissions are also relatively poorly understood (Liang et al. 2014). We update the time series for CTC as follows. First, we update the lifetime of CTC to match the 2022 SAOD assumed lifetime of 30 years, and we then re-calculate all future emissions based on the same approaches in each of the respective SAODs. This adjustment impacts future projected mole fractions of

CTC due to the rate of atmospheric decline and also impacts observationally-derived emissions that are used to inform the projected emissions. Next, we update the timeseries of observed mole fractions out to 2021 from the 2022 SAOD, as is done in Step 4, to reflect how the original projection methods from each SAOD would be impacted by the actual observed mole

fractions.    The final part of the modified CTC projection involves updating the future emissions projection method to what was used in the 2022 SAOD, which brings both historical and future CTC mole fraction time series to the timeseries used in the 2022 SAOD.  We refer to this final part as accounting for future feedstock emissions, though the 15 Gg/year of continued emission was meant to comprise all potential emissions, including those that might arise from unreported production.

*Step 7: Update all other gases to WMO (2022) values*

For the final step in quantifying contributions to expected delays in EESC, we update the time series for the abundance of the remaining 12 ODSs to the values given in the 2022 SAOD.

## 4 Results and Discussion

The various formulations of EESC used in the past five SAOD reports have resulted in substantial differences in the 1980 return date reported in these assessments (Fig. 3).  There are relatively large differences in the magnitude of EESC given in the 2006 SAOD compared to all subsequent SAOD reports (Fig. 3a). The larger EESC magnitude in the 2006 report is due to their use of a single set of FRFs for each ODS representative of the global stratosphere given in Table 8-1 of the 2006 SAOD, rather than the adoption of separate sets of FRFs for the mid-latitude lower stratosphere (that is, 3-year-old air) and the polar stratosphere (5.5-year-old air), which commenced with the 2010 SAOD. Hence, the peak value of EESC given in the 2006 report falls in between the peaks of EESC for 3-year-old air and 5.5-year-old air given in the 2010 report. However, EESC return dates as reported in the assessments do not meaningfully change between the 2006 SAOD report and the subsequent three SAODs.  Mid-latitude EESC values for the 2006 to 2018 SAOD reports yield a return to 1980 date between 2046 and 2050.  In contrast, the 2022 SAOD report provides an estimated return date of 2066.  This disparity is due in part to the use of the Engel et al. (2018) formulation of EESC in the 2022 report, which effectively uses a new formulation that accounts for the interaction of tropospheric trends in the organic species with atmospheric loss of these compounds (Ostermöller et al. 2017; see also Section 2.1 and Box 1-4 of the 2018 SAOD report). Use of the Engel et al. (2018) approach in the 2022 SAOD results in a rightward shift in the EESC timeseries that reduces the value of EESC in 1980, consequently delaying 1980 return date for EESC relative to all of the previous assessments (Fig. 3a).  Various other counteracting changes in the formulation of EESC for the 2006 to 2018 SAOD reports resulted in near constancy of the return to 1980 level for EESC. With each subsequent SAOD report, there was a tendency for the surface mixing ratios of ODSs, except for HCFC-22, to return to their 1980 levels at later dates, which all else being equal would have led to incremental delays in the return to 1980 date of EESC.  However, also with each subsequent SAOD report, there were incremental changes in the approach used to compute EESC that largely counteracted these incremental delays.  It is therefore instructive to examine the return to 1980 EESC levels for the assessed time series of the 16 principal ODSs of each SAOD report using an identical formulation for EESC.

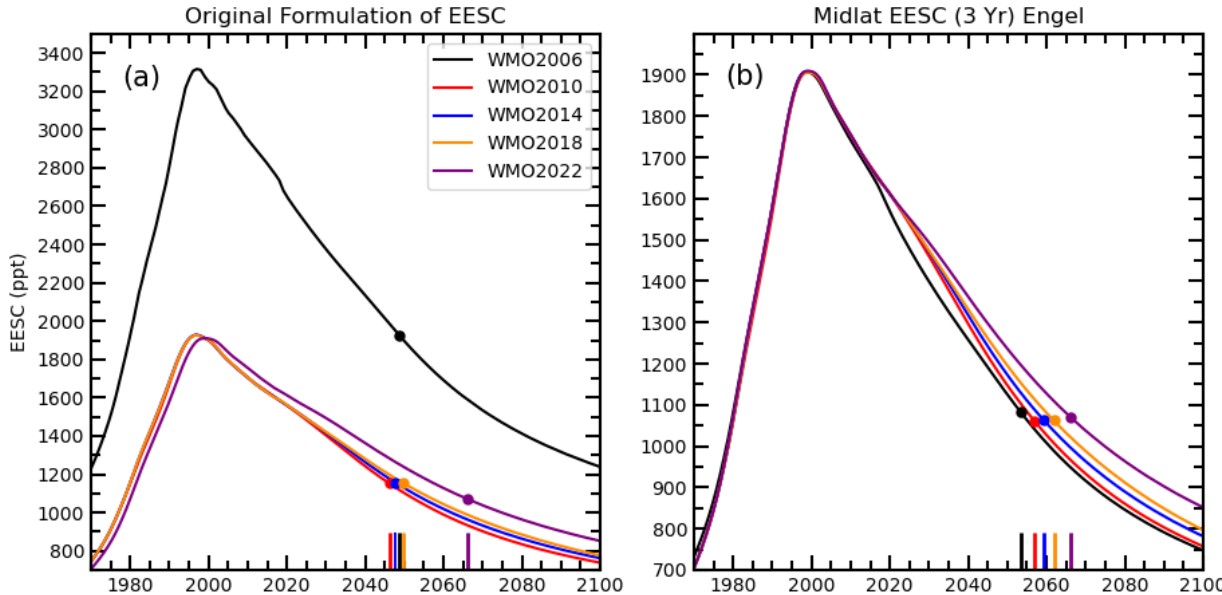

**Figure 3.** EESC calculations applied to each WMO assessment atmospheric mole fraction table of the 16 primary ODSs. (a) EESC formulation as it appears in the original assessments. (b) Engel et al. (2018) EESC calculation applied to the atmospheric mole fractions given in each assessment. Dots and vertical lines on the x-axis refer to the first month in which EESC returns to below 1980 levels for the respective WMO assessment.

Using the Engel et al. (2018) formulation for EESC applied to the time series of ODS mole fractions given in the past five SAOD reports, we see a near consistent delay in the return to 1980 EESC levels between reports (Fig. 3b). For each consecutive SAOD, the return to 1980 date lags that given in the prior report by about 2 to 4 years. Further, using the identical formulation for EESC (Engel et al. 2018) shortens the difference in the 1980 return date between the 2006 and 2022 SAOD reports from 17 to 12.6 years, which we investigate below. Note that the value of EESC in 1980 (that is, the return to 1980 target) does not perfectly align, despite the use of an identical formulation for the calculation of EESC (Fig. 3b). These slight shifts are a result of updating historical atmospheric mole fractions of ODSs between assessments, with the most significant change arising from methyl bromide (Table 2).

**Table 2:** Independent incremental change (in yrs) between assessments by gas for mid-latitude EESC return date using the Engel et al. (2018) formulation. Gases are ordered by total contribution to change in EESC return date between 2006 and 2022, where the 2022 SAOD's estimated EESC return date is 2066.0. For each gas' calculation, the subsequent SAOD's

timeseries is adopted for that gas, while the 15 other gases maintain the timeseries from the original SAOD. The difference is then calculated relative to the original SAOD's timeseries for all 16 gases.

| | WMO 2006 | WMO 2010 | WMO 2014 | WMO 2018 |
|---|---|---|---|---|
| Initial EESC return date Engel et al. (2018) | 2053.5 | 2056.9 | 2059.4 | 2061.9 |
| | Difference from 2006 to 2010 (yrs) | Difference from 2010 to 2014 (yrs) | Difference from 2014 to 2018 (yrs) | Difference from 2018 to 2022 (yrs) |
| CFC-11 (CCl$_3$F) | 0.55 | 1.82 | 0.15 | 2.42 |
| Carbon tetrachloride (CCl4) | 1.76 | 0.67 | 3.3 | -0.83 |
| Halon 1301 (CBrF$_3$) | 0.53 | 0.2 | 0.04 | 0.6 |
| CFC-12 (CCl$_2$F$_2$) | 0.02 | 0.12 | 0.04 | 0.64 |
| Methyl bromide (CH$_3$Br) | 2.15* | -0.43 | -1.02 | 0.01 |
| CFC-113 (CCl$_2$FCClF$_2$) | 0.06 | 0.17 | 0.03 | 0.21 |
| Halon 2402 (CBrF$_2$CBrF$_2$) | 0.27 | 0.53 | -0.02 | -0.43 |
| HCFC-142b (CH$_3$CClF$_2$) | 0.16 | -0.03 | 0 | 0.09 |
| Methyl chloroform (CH$_3$CCl$_3$) | 0.02 | 0 | 0 | 0.06 |
| CFC-114 (CClF$_2$CClF$_2$) | -0.01 | 0 | 0.01 | 0.05 |
| CFC-115 (CClF$_2$CClF$_3$) | 0 | 0 | 0 | 0 |
| Halon 1202 (CBr$_2$F$_2$) | 0 | -0.01 | 0 | 0 |
| Halon 1211 (CBrClF$_2$) | -0.31 | -0.07 | -0.01 | 0.37 |
| HCFC-141b (CH$_3$CCl$_2$F) | 0 | -0.09 | -0.02 | 0.03 |
| Methyl chloride (CH$_3$Cl) | 0.03 | -0.43 | 0 | 0 |
| HCFC-22 (CHF$_2$Cl) | -1.89 | -0.05 | 0.06 | 0.74 |
| Sum | 3.34 | 2.40 | 2.56 | 3.96 |

*Updating EESC values from changes in methyl bromide between WMO (2006) and WMO (2010) leads to a relatively large decrease in the value of EESC in 1980, from 1082 ppt to 1060 ppt.

The role of the choice of EESC formulation in delaying return dates is further explored in Fig. 4. The SAOD reports provide estimates of EESC for both midlatitude (assumed to be 3-year-old air) and polar (5.5-year-old air) stratospheric regions, which are used in many papers (and the assessments) as proxies for the recovery of midlatitude and polar ozone to perturbations caused by anthropogenic halogens. When normalizing the original formulations of EESC to the value in 1980 reported in those assessments, the 2022 SAOD report is a notable outlier for EESC of midlatitude regions (Fig. 4a).

Identical formulations of EESC, using either the Engel et al. (2018), Newman et al. (2007), or Daniel et al. (1995) method, result in consistent delays in the return dates for midlatitude EESC to 1980 levels with each subsequent SAOD report (Fig. 4b and 4c, Figure S1). The return to 1980 date for midlatitude EESC is delayed by 12.6, 9.8, and 10.4 years when switching from the ODS mole fraction tables given in the 2006 SAOD report to the 2022 report upon use of either the Engel et al. (2018), Newman et al. (2007), and Daniel et al. (1995) methods, respectively. Therefore, the mole fraction tables for ODSs given over the 16-year assessment period that are central to this study have played a key role in the delay in return of stratospheric halogens to the 1980 level, regardless of which approach is used to compute EESC.

There are also large differences in the return to 1980 dates *between* the various formulations for EESC. For example, return dates of midlatitude EESC found using Engel et al. (2018) lag those of Newman et al. (2007) by 13.8 and 3.5 years for midlatitude and polar air, respectively, when using the ODS mole fraction table given in the 2022 SOAD report. The later return date of the Engel et al. (2018) formulations is largely driven by their use of a method that accounts for the relationship of tropospheric source gas trends and stratospheric chemical breakdown. Their EESC formulation takes into account the time needed to release the halogens from their source gases. The inorganic fraction, which EESC represents, is therefore weighted towards longer transit times and thus lags the troposphere more strongly than in the older formulation. The Engel et al. (2018) approach used in the 2022 SAOD report again leads to lower EESC values during the ascending phase of the tropospheric halogen loading and higher EESC values during the descending (recovery) phase of tropospheric halogen loading, and thus to a longer time frame needed to reach 1980 EESC values. Applying the Engel et al. (2018) formulation for EESC to the ODS mole fraction table of the 2010 SAOD delays the return to 1980 value for midlatitude air by about 3.5 years compared to the return date found using the mole fractions from the 2006 SAOD. Relative to the 2010 SAOD report, the 1980 return date is delayed by 2.4 years when using the mole fraction table of the 2014 report (Figure 4b). The 2018 SAOD report exhibits another 2.5-year delay relative to the 2014 report. The 2022 SAOD report was the first report to adopt the Engel et al. (2018) formulation for EESC as the primary approach and showed an additional 4.1-year delay in the return date relative to the 2018 SAOD with the Engel et al. (2018) approach employed for both sets of ODS mole fractions. The top panels of Figure 4 illustrate the important roles that ODS mole fraction changes and alternate formulations of EESC have played in delaying the return of EESC to the 1980 level for mid-latitude air, over the past 16-year assessment period.

While the Newman et al. (2007) and Engel et al. (2018) formulations result in large differences in EESC for mid-latitude regions, these two formulations result in similar EESC return dates for polar regions (Fig. 4 e and f). This difference is due to the Engel et al. (2018) EESC formulation attempting to weight the age of air by the timing of source gas dissociation. This leads to differences in estimated mole fractions of active (inorganic) halogens in mid-latitudes, as photolysis largely occurs in the tropical stratosphere and affects some ODSs more than others. The formulations give similar estimates of EESC for polar

regions because the transit through the stratosphere from injection (in the tropics) to polar descent includes the transit of air parcels through the upper branch of the Brewer-Dobson circulation. This results in nearly complete loss of most ODSs, due to longer residence time in the stratosphere and most importantly exposure to a more intense ultraviolet radiation environment than is seen for most 3-year old, mid-latitude air parcels. Thus, the age of air associated with dissociated ODSs is much more similar to the age of an inert tracer in polar regions than is the case for mid-latitude air.


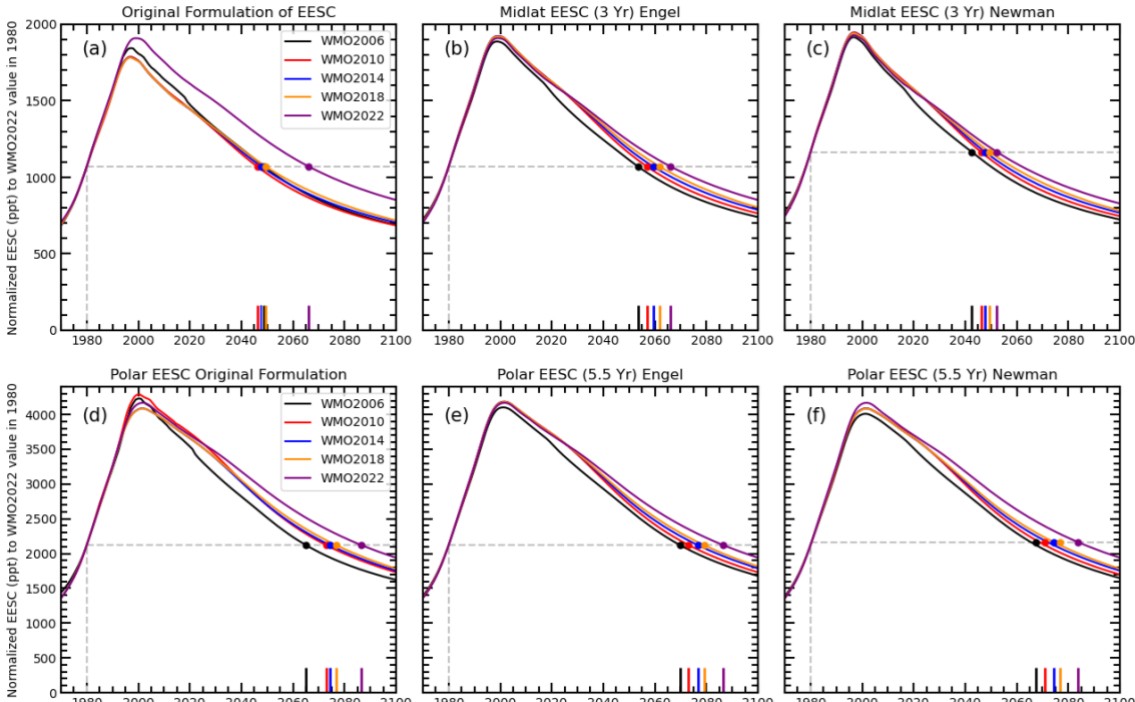

**Figure 4.** EESC calculations applied to each WMO assessment atmospheric mole fractions of the 16 primary ODSs, normalized to the value of EESC in 1980 reported by WMO (2022). (a) The original assessment's mid-latitude formulation of EESC formulation. (b) mid-latitude EESC estimates for each WMO assessment's atmospheric mole fractions using the

Engel et al. (2018) formulation. (c) as in (b) but using the Newman et al. (2007) formulation. (d)-(f) as in (a)-(c) but for polar EESC calculations. Dots and vertical lines on the x-axis refer to the first month in which EESC returns to below 1980 levels for the respective WMO assessment.


Table 2 provides the calculated contribution of each ODS to the delay in the return to 1980 EESC values. CFC-11 and CTC account for the largest delays in EESC recovery between the 2006 and 2022 SAOD reports, each delaying cumulative recovery over this period by ~ 4.9 years (the ~4.9 year delay is the sum of the four individual delays given for both CFC-11 and CTC in the table). Halon-1301 and CFC-12 contribute the third and fourth most substantial delays of 1.4 and 0.8 years, respectively.

Note that the impacts on the 1980 return date shown in Table 2 are calculated for each ODS independently. The return to 1980 date for EESC is affected by past and future changes of all ODSs in a small, non-linear manner due to the nonlinearity of the EESC time series, such that the sum of the independent impacts across all ODSs (Table 2), when all are changed simultaneously, is not precisely equal to the cumulative sum of the impacts on return date when all ODSs are changed individually (Table S2).


Of the four ODSs contributing to the largest delays in ozone recovery, CFC-11, CFC-12 and halon-1301 are subject to significant banking. We further investigate the role of each change in modeling assumptions for these three ODSs as well as the other key factors in the changing return dates in Fig. 5. Since 2006, projected mole fractions of HCFC-22 have decreased by a greater amount than had once been projected in the 2006 SAOD. However, the effect on EESC of a faster decline in the

atmospheric abundance of HCFC-22 than had once been forecast has been offset by higher than expected atmospheric abundances of the other aggregated 11 ODSs (that is, the other 15 ODSs excluding the top 4 [CFC-11, CTC, halon-1301, and CFC-12]). The implications of HCFC-22 relative to other ODSs are further explored in Fig. 6. Between the 2006 SAOD and 2010 SAOD reports, updates in methyl bromide delayed the EESC return date by ~2 years. This change is a result of updates in pre-1980 methyl bromide mole fractions, which lowered the 1980 EESC baseline value between the 2006 and 2010

assessments (Fig. 3b).

The primary results of this analysis are shown in Fig. 5 and Table S2, which quantify the contribution of each modeling assumption to the delay in EESC return dates that is found for midlatitude air for the past 5 SAOD reports. Updating the atmospheric lifetime for CFC-11 from the 45-year value used in the 2006 and 2010 SAOD reports to the 52-year value used

in the 2014 to 2022 reports results in a 2.2-year delay in EESC return date and is thus a key single factor. This delay is due to the projected atmospheric mole fractions declining more quickly in the earlier assessments with the shorter assumed CFC-11 lifetime, and also a result of the impact of lifetimes on earlier bank estimation methods. In the 2010 SAOD, for example, the shorter lifetime leads to higher inferred emissions during the time when atmospheric mole fraction observations were available, and because production was fixed, these higher emissions were modeled as emissions from banks. Therefore, banks were

estimated to deplete more quickly when using the lower atmospheric lifetime.

Updates in observed atmospheric mole fractions do not substantially impact our EESC return date estimates for banked gases, relative to the other factors. One initially counter-intuitive result is the impact of observed mole fractions of CFC-11 on the 1980 return date, for each SAOD report from 2006 to 2018 (Fig. 5). When observed mole fractions were higher than had been expected, this finding accelerated the estimated return date using each SAOD's respective projection method. The higher mole fractions were assumed to be due to higher-than-expected emissions from banks, which was achieved by increasing the release fractions from banks. This assumption led to estimated bank values decreasing more quickly as the date of each SAOD report advanced, thus being a smaller source of future emission which in turn moved up the return date for EESC. This result is in part due to the assumption in SAODs from 2006 to 2018 that the global production of these banked gases was well known, and that the uncertain parameters controlling bottom-up emissions were in bank release rates, not industrial production. This assumption regarding highly certain production values was relaxed in the bank modeling approach used in 2022 SAOD report, which has led to bank release rates being less sensitive to observed mole fractions.

Updating the bank methodology to the 2022 SAOD report results in notable delays in EESC return dates for all three gases, though most substantially for CFC-11. The 2022 SAOD methodology results in significantly larger bank estimates compared to the prior assessments (Fig. 1), primarily driven by allowing for uncertainty in the values of production of ODSs. By relaxing the assumption of completely accurate production reporting under the Montreal Protocol and even full compliance under the Protocol, higher atmospheric mole fractions and inferred emissions of ODSs may result in higher posterior production estimates, which in turn accumulate into higher bank estimates. This was the case with unexpected emissions of CFC-11 after 2012 (Benish et al., 2021; Lickley et al., 2022; Montzka et al., 2018; Park et al., 2018; Rigby et al., 2019), where allowing for production uncertainty in the banks modeling framework over this time period resulted in posterior production estimates that were non-zero following 2010, and larger posterior bank values than what was inferred by assuming zero production uncertainty and full compliance with the Protocol following 2010.

Changes in understanding of the processes impacting CTC have led to nearly consistent delays in expected EESC return dates (Fig. 5). Updating the atmospheric lifetime from 26 years, which was used in the 2006, 2010, and 2014 SAOD reports to the 30-year value used in the 2022 SAOD had two competing effects on EESC return date. The longer lifetime results in lower inferred emissions during the time atmospheric mole fraction observations were available; however, it also implies a slower decay of the gas in the atmosphere, resulting in a 0.9 year delay in EESC return date relative to the 2006 SAOD. Likewise, the higher-than-projected observed mole fractions of CTC compared with earlier projections contributed an additional 1-year delay in the EESC return date. Because CTC is not a banked chemical, there is no compensating emission source to explain these higher-than-expected mole fractions in earlier SAODs, so the higher-than-expected inferred emissions would be modeled strictly as an emission source from new production of CTC. Finally, the additional assumption in the 2022 SAOD report of a long-term, continuous feedstock emissions of 15 Gg/year, which was not accounted for in the projections used in any of the earlier SAODs (Fig. 2), contributes an addition 3.1 years to the delay in EESC compared to the 2006 SAOD report. Note,

however, that this represents an acceleration in EESC return date relative the 2018 SAOD, where the emissions decay function resulted in larger CTC emissions in the first half of the 21ˢᵗ century relative to the 2022 projections (Figure 2). Overall, updated modeling assumptions for CTC between the 2006 and 2022 SAODs delays the expected EESC return date by a total of 4.9 years.

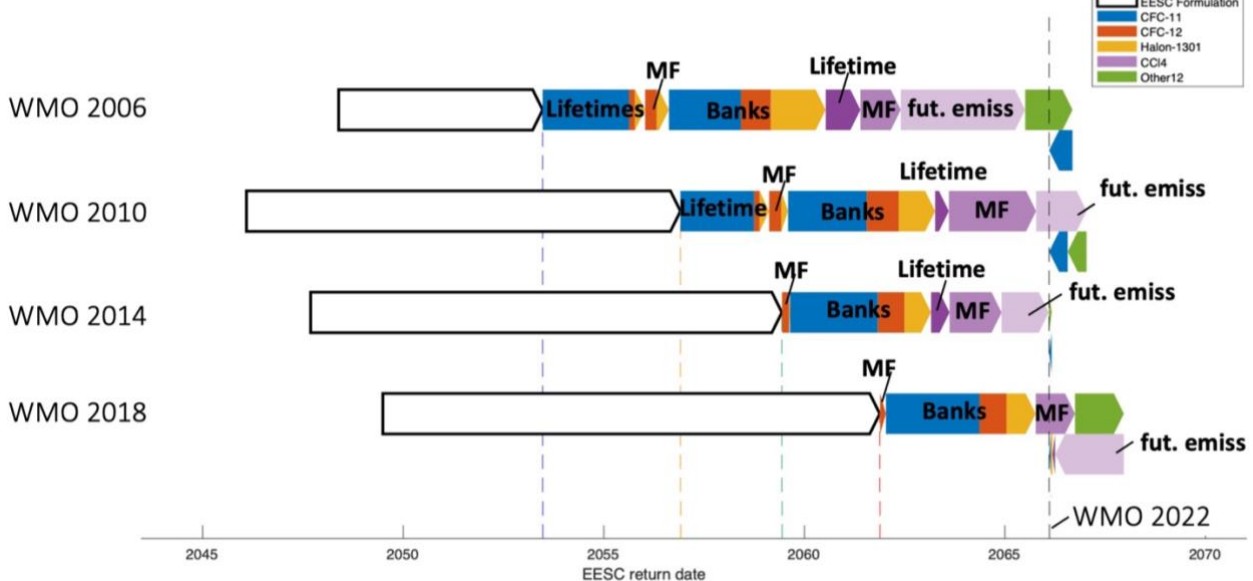


**Figure 5.** EESC return dates to 1980 levels for each sequential update to the original assessment's methods. The vertical dashed lines correspond to the return date that is estimated after applying the identical Engel et al. (2018) EESC formulation to each of the original WMO atmospheric mole fraction time series for the 16 primary ODSs. The corresponding change resulting from the EESC formulation update is shown by the white arrows, followed by atmospheric lifetime assumptions for
the three most prominent bank gases responsible for delaying EESC return date (CFC-11, CFC-12, and halon-1301). The next update corresponds to the update in observed mole fractions for these three gases since publication of each assessment to those measurements used in the 2022 SAODS, indicated by MF, followed by the banks update. The contribution of projections of carbon tetrachloride (CTC) to delaying EESC return dates is shown in purple for the lifetime updates (dark purple), the observed mole fraction updates (medium purple), and the future feedstock emissions projections (light purple). The remaining
12 gases are shown in green. If the update corresponds to a delay in EESC return date, then the arrow points to the right. If updated assumptions accelerate the return date, the arrow points to the left.

Much of this work focuses on ODSs and their role in estimating EESC, which is an estimate of inorganic halogens in the stratosphere. Because stratospheric halogens originate from organic compounds that are accurately measured in the
troposphere, the quantity effective equivalent chlorine (EECl) (Montzka et al. 1996) has been considered in parallel with EESC when reporting trends in ODSs. Time series of EECl have been highlighted in Figure 1 of the Executive Summary of the 2018

and 2022 SAOD reports, though an unconventional definition of EECl was used, which did not account for FRFs in the calculation. Here, we term this alternative definition equivalent tropospheric chlorine (ETCl). Figure 6a shows historical and projected values of ETCl, found using the time series for 16 principal ODSs of each SAOD report considered in this analysis (Fig. 6a). The 1980 return date of ETCl for SAODs from 2006 to 2018 are all clustered together around 2046/2047, whereas for the 2022 SAOD report the 1980 return date of ETCl is around 2054. Repeating this calculation but weighting each ODS using FRFs, following the definition of Montzka et al. (1996), we find that the EECl return dates are more evenly delayed between various SAODs, more notably in the midlatitudes than for polar regions (Figure 6b, c). The reason for this difference is that ETCl assumes FRFs of unity for all species, resulting in a larger weighting for the HCFCs than is found for either EECl or EESC. Between the 2006 and 2010 SAODs, the projected emissions of HCFC-22 dropped by a factor of ~2, reflecting the 2007 decision by the Parties to accelerate the phase-out of HCFCs (Montzka et al., 2015). For ETCl, this earlier phase-out of HCFC-22 offsets the delay in the return to 1980 value due to projected slower declines in other ODSs. While the drop in HCFC-22 makes a large contribution to ETCl, there is a substantially smaller effect on EECl (Figures 6b,c) and EESC (Figure 4) because of the use of FRFs for HCFC-22 of 0.15 and 0.44 for mid-latitude and polar air, respectively, compared to the use of FRF equal to unity for HCFC-22 and all other species in the formulation of ETCl. A proper accounting of FRFs is needed, as is done in EECl and EESC but not in ETCl, should past and projected tropospheric abundances of ODSs be used as a proxy for how future ozone depletion will be affected by anthropogenic halogens. Finally, the much longer return to 1980 dates of EESC (Figure 4) compared to EECl (Figure 6b,c) is caused by the time it takes for ODSs to reach various levels of the stratosphere as well as the distribution of these times, as is included in the age-of-air spectrum inherent in the definition of EESC.

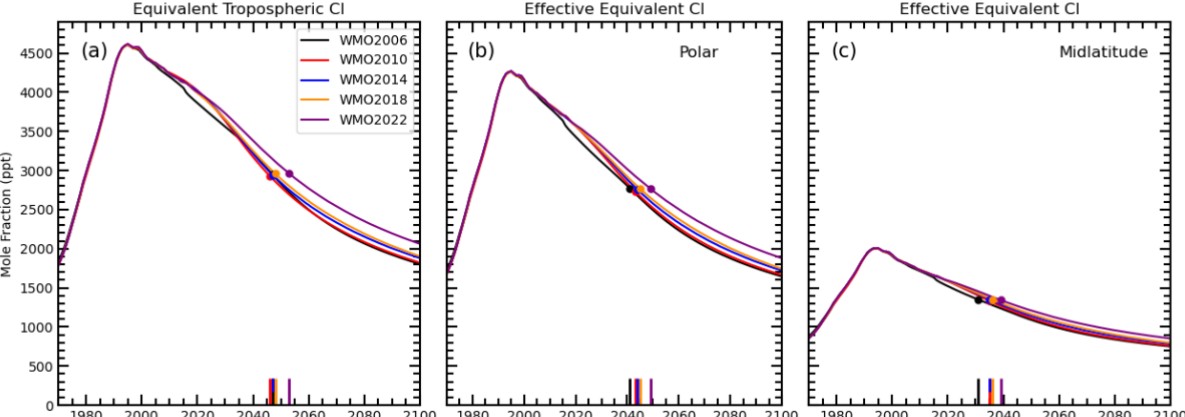

**Figure 6.** Equivalent tropospheric chlorine (ETCl) and effective equivalent chlorine (EECl) estimates for each WMO assessment, using atmospheric mole fractions of the 16 primary ODSs. (a) ETCl, computed in the same unconventional manner as the quantity termed "Equivalent effective chlorine" that was shown in figure ES-1 of the 2018 and 2022 SAOD Executive Summaries; (b) EECl calculated following (Montzka et al. 1996) using polar time-independent fractional release factors from

Table 2 of (Engel et al. 2018); and (c) EECl calculated following (Montzka et al. 1996) using midlatitude time-independent fractional release factors from Table 1 of (Engel et al. 2018).


## 5 Conclusions

Between the 2006 and the 2022 SAOD reports, the expected year for which EESC of midlatitude (3-year old air) returns to the value of EESC at the start of 1980 has been delayed by ~17 years. This change suggests, all else being equal, an approximate

17-year delay in the recovery of midlatitude ozone column with respect to the value that occurred in 1980. Reported EESC recovery dates were relatively consistent between the 2006 and 2018 SAOD reports, with the value given in the 2022 report appearing to be an outlier, though this was in part to be expected from the adoption of a new formulation for EESC (Engel et al. 2018). However, applying identical formulations to the computation of EESC to projections of the 16 principal ODSs of each SAOD report indicates, as shown above, that the EESC recovery time has been consistently delayed by 2 to 4 years

between each successive SAOD report. Thus, the changing formulation of EESC between the 2006 and 2018 SAOD reports has obscured the fact that the assessed projections of the atmospheric abundances of ODSs as a whole have been consistently updated to higher values, on average, between consecutive reports. Applying an identical formulation for EESC (Engel et al., 2018) to the projections of ODSs from each SAOD report results in a delay of 12.6 years, between the 2006 and 2022 SAOD reports, for the recovery of EESC of midlatitude air to the value found for the start of 1980. Lifetime assumptions in the 2006

and 2010 assessments for CFC-11 and other key ODSs were lower than the current best estimates, contributing to an earlier expected return date for EESC than found using lifetimes in the 2022 SAOD report. Since the 2006 SAOD report, changes in atmospheric lifetime estimates can explain approximately ~3.5 years of the difference between the 2006 and 2022 SAOD projected return dates. Higher than expected mole fractions of ODSs explain ~1 year of the difference, largely due to observed mole fractions of CTC, which contributed a higher baseline and slower rate of decline in future emissions projections. Changes

in bank estimates account for another ~4 years of the difference in EESC return date, and updated future emissions projections of CTC, largely due to assumed continued feedstock emissions, account for ~3 years of the difference. The remaining 12 ODS mole fraction projection updates account for an additional net change of ~1 year between SAODs.

An important update in the 2022 SAOD pertains to the assumptions that historical production of ODSs were in compliance

with the Montreal Protocol and that reported production numbers were fully accurate. In the baseline scenarios of earlier SAODs, it was assumed that there was no unreported production and therefore unexpected emissions were accounted for by higher release rates from banks. For the 2022 SAOD report, new production of controlled substances not in compliance with the Montreal Protocol was considered; this new production is included implicitly through increases to both atmospheric mole fractions and explicitly through the bank size of the affected ODS. These updates have been made in light of evidence of

unreported production of halocarbons in recent years (Benish et al., 2021; Lickley et al., 2022; Montzka et al., 2018; Park et al., 2018; Rigby et al., 2019; Montzka et al., 2021; Park et al., 2021; Sherry et al., 2018) as well as during the 1980s in the Soviet Union at an amount that accounted for ~20% of global production of CFC-11 (Gamlen et al., 1986), suggesting historical production may have been consistently underestimated in earlier SAOD reports.  Production projections of CTC have similarly been consistently underestimated in the SAOD reports (SPARC, 2016). The CTC budget continues to be a source of

uncertainty, as observationally-derived emissions are consistently higher than bottom-up estimates (Daniel and Reimann, 2022). Recent studies have made progress on budget closure  (Sherry et al. 2018; Liang et al. 2014; Park et al. 2018), though bottom-up sources of 15-25 Gg/year (Sherry et al., 2018) are still not within the top-down global emissions range of 34 - 45 Gg/year (Liang et al. 2014).  Li et al. (2024) found current atmospheric emissions of CTC from numerous industrial sources such as the manufacture of general-purpose machinery, raw chemical materials, and chemical products. Currently, the use of

CTC and other ODSs used as feedstock in the manufacturing process is exempt from control under the Montreal Protocol, likely due to an assumption that the associated atmospheric releases would remain small.  The findings of Li et al. (2024), coupled with the continued tendency of atmospheric mole fractions of CTC to lie above prior projections, suggest a portion of the slower than expected decline of EESC since 2006 is caused by inadvertent atmospheric releases of CTC from a wide range of industrial activities.


Following the publication of the 2022 SAOD, further evidence has emerged of increasing mole fractions of CFCs from 2010 to 2020 (Western et al. 2023), thought to be driven in part by feedstock-related emissions, and reports have emerged of unreported feedstock emissions at chemical plants (EIA 2023).  The apparent leakage from feedstock activity may warrant increasing controls on their production processes (Andersen et al., 2021; Lickley et al., 2021).  Banks represent another

opportunity for reducing future halocarbon emissions. While the CFC-11 bank resides largely in foams, which is difficult to recover, CFC-12 used in refrigeration, and the use of halon-1301 as a fire suppressant may be more accessible for bank collection and subsequent destruction. Full recovery of CFC-12 and halon-1301 banks would accelerate estimated ozone recovery by ~ 3 years, with total bank collection representing an opportunity for accelerating ozone recovery ~6 years of delay (Lickley et al. 2020, 2022).  Unexpected emissions and additional controls on ODSs, such as the ones described above, would

all impact estimates of the return to 1980 date for EESC that will be given in future assessments.

In addition to changes in controls on feedstock and bank emissions, updates in the representation of atmospheric processes may also result in changes in expected EESC recovery time in future SAOD reports.  Future updates may include changes in estimated ODS atmospheric residence times resulting, for example, from anthropogenic global warming driven changes to the

Brewer-Dobson circulation (Prather et al., 2023; Fleming et al. 2011), ocean exchanges, and changes in the hydroxyl radical (Wang et al. 2023).  There is emerging evidence that very short-lived (VSL) chlorine compounds, which are largely anthropogenic and are not controlled by the Montreal Protocol, might be responsible for the slower than expected decline of

HCl in the lower stratosphere (Bednarz et al., 2022; Hossaini et al., 2015; Hossaini et al., 2019). If so, then VSL chlorine compounds might need to be considered in future formulations of EESC.


EESC baseline projections serve two important purposes for policy makers. First, they are designed to reflect how the current controls in place under the Montreal Protocol are expected to impact stratospheric halogens, and hence the recovery of the ozone layer. Parties can use this information to identify which additional restrictions could potentially be considered for safeguarding the ozone layer and climate system. Second, the baseline mole fraction projections used to calculate EESC sets

expectations with regards to future abundances of ODSs. These projections have been proven valuable in identifying new and illicit production of banned substances in breach of the Protocol (e.g. Montzka et al. 2018). While global compliance has not been absolute, the effectiveness of the Protocol is clearly evidenced by the current declines in the value of EESC along with initial signs of the recovery of the ozone layer (Solomon et al., 2016; Dhomse et al., 2018; Weber et al., 2022). The present study shows that consistent delays in the estimated EESC return date to the 1980 level is partially due to unreported production

of banned ODSs, and partially driven by scientific uncertainty in atmospheric lifetimes and estimates of bank sizes. We would expect updates in the modeling of atmospheric processes to affect expected EESC return dates in the future, and it could be valuable for future SAOD reports to consider including uncertainty quantifications in their baseline projections to account for the uncertainty in current scientific understanding (for example the uncertainty of atmospheric lifetimes of ODSs). However, we would expect that each new update in the representation of atmospheric processes could lead to either accelerations or

delays in expected EESC return times. Changes in expected EESC return dates resulting mainly in delays are not expected to be likely a result of atmospheric uncertainties alone, but rather may suggest the potential for either breaches in the Protocol or significant emissions resulting from the use of ODSs as feedstock, which are not controlled by the Protocol. A continued trend of delayed EESC return dates in future SAODs would suggest careful consideration is warranted regarding current reporting and monitoring procedures and regarding our understanding of ODS lifetimes and how to best characterize emissions over

time.

**Author Contributions**

All authors designed the methodology. M.J.L., J.S.D, and R.J.S. conducted the analysis. M.J.L and L.A.M. created the figures. All authors contributed to the writing and editing of the manuscript.

**Data Availability**

All data used in this study is available through a public repository (https://doi.org/10.5281/zenodo.13952811).


**Code Availability**

All code used in this study is available through a public repository (https://doi.org/10.5281/zenodo.13952811).

## Competing Interests

The authors declare that they have no conflict of interest.

## Acknowledgements

M.J.L. would like to acknowledge support from the Atmospheric Chemistry Division of the National Science Foundation (grant no. 2128617) and from VoLo Foundation. R.J.S. and L.M. would like to acknowledge support from NASA Atmospheric

Composition and Modeling Program (Grant No. 80NSSC19K098).

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
