# Peer review of "The return to 1980 stratospheric halogen levels: A moving target in ozone assessments from 2006 to 2022"

_EGUsphere, 2024_

## Author Response (AR1)

RC1:

The manuscript by Lickley et al. discusses the projected return date of EESC to 1980 levels, which is generally taken as a proxy for an expected recovery of the ozone layer. Most importantly, this measure has been used regularly in the WMO SAOD reports. The paper is well written, scientifically sound and will be an important contribution to explain why EESC recovery dates have constantly shifted further into the future. This is a very important contribution and clearly differentiates the different drivers. The approach to use a consistent EESC formulation is a very important and I think that the manuscript is nearly ready for publication. There are some minor points which I would like the authors to consider.

Thank you for this positive feedback!

Minor comments:

l. 58: variable or changing atmospheric transport has also been discussed as a possible reason for the delayed decrease of CFC-11 mole fractions.

Thank you, we have added this explanation to our list (Line 66):

"variability in atmospheric transport (Ray et al. 2020)"

l. 124: rho is the in this case the mole fraction which would be expected in absence of chemical loss.

We have made this correction.

l. 133: if a is the fractional release factor of CFC-11, then the fi values in (1) must be relative values to CFC-11 (see formula (1) in box 8.1. of WMO 2006.

No change in response to this comment, because the definition of fi was stated as being relative to CFC-11 on lines 123 and 124 of the original submission.

l. 158: modeled concentrations (not concentrations modeled). And actually these are mole fractions, rather than concentrations.

Thank.  We've made this correction throughout.

l. 187: form many species modelled lifetimes were used in the SPARC 2013 assessment.

This is a good point.  We've added a clarifying sentence (Line 322-323):

"However, for many species, modelled lifetimes alone inform the atmospheric lifetimes used in the SAODs (as reported in SPARC, 2013)."

l. 248: I could not find a shaded region in the plot in my print-out.

This was an issue in the conversion of our figure into a pdf format, which has been corrected.

l. 353: as above: if relative fractional release to the fractional release of CFC-11 is used, then the whole calculation should be multiplied by the frf of CFC-11 (again, see box 8.1. in WMO 2006). It seems to me that the calculation in WMO 2006 and in the plots in chapter 8 of WMO 2006 may actually be missing this. Maybe it might be useful to just apply this factor here, so that the calculation is consistent with other reports?

Thank you for pointing this out. We believe we made an error in our initial interpretation of the WMO 2006 calculation and have corrected it now to be consistent with the report. We've updated the text accordingly (Line 560 - 564):

"The larger EESC magnitude in the 2006 report is due to their use of a single set of FRFs for each ODS representative of the global stratosphere given in Table 8-1 of the 2006 SAOD, rather than the adoption of separate sets of FRFs for the mid-latitude lower stratosphere (that is, 3-year-old air) and the polar stratosphere (5.5-year-old air), which commenced with the 2010 SAOD. Hence, the peak value of EESC given in the 2006 report falls in between the peaks of EESC for 3-year-old air and 5.5-year-old air given in the 2010 report."

l. 357: you may want to specify that this is the mid latitude return date.

Thank you for catching this, we've added this correction.

L 401: you may want to add that this large difference is mainly due to a different value of EESC derived for 1980 with my formulation, which takes into account that the released fraction has on average experienced longer transit times, thus that the inorganic chlorine is dominated by an input which is "older" than the mean age value.

Thank you for this suggestion. We've added an explanation here (Line 722-729):
"The later return date of the Engel et al. (2018) formulations is largely driven by their use of a method that accounts for the relationship of tropospheric source gas trends and stratospheric chemical breakdown. Their EESC formulation takes into account the time needed to release the halogens from their source gases. The inorganic fraction, which EESC represents, is therefore weighted towards longer transit times and thus lags the troposphere more strongly than in the older formulation. The Engel et al. (2018) approach used in the 2022 SAOD report again leads to lower EESC values during the ascending phase of the tropospheric halogen loading and

higher EESC values during the descending (recovery) phase of tropospheric halogen loading, and thus to a longer time frame needed to reach 1980 EESC values."

l. 415 (Figure 4): maybe you could add a line going upwards from 1980 and then parallel in order to visualize the calculation of return date.

Thank you for this suggestion, we've added an updated figure to the text to provide this illustration.

l. 427: I'm not sure I understand this. EESC is (as shown in your formula (1) a sum, so the effect of individual compound should add up linearly. Or have I misunderstood the statement?

Yes, EESC is a linear sum of these values. It is the "return to 1980 EESC levels" that is non-linear due to the exponential decay function, however, the assumption of linearity when taking the combined contributions from each ODS results is only a small discrepancy. We've added some small edits to the text to clarify (Line 1048 – 1052):

"The return to 1980 date for EESC is affected by past and future changes of all ODSs in a small, non-linear manner due to the nonlinearity of the EESC time series, such that the sum of the independent impacts across all ODSs (Table 2), when all are changed simultaneously, is not precisely equal to the cumulative sum of the impacts on return date when all ODSs are changed individually (Table S2)."

l. 435: are these increases or slower than expected decreases?

Good point. This is higher than expected abundances. We've made this correction to the text.

l. 447: I find the term bank emissions difficult; maybe use emissions from banks?

We've changed the wording to emissions from banks.

l. 498 and following: I'm not sure if the discussion of EECL is very important here or adds much more to the points you make.

We feel this section is essential, given the reliance on EECL in the Executive Summary of so many SAOD reports (e.g. Fig 1 in 2018 and 2022 Executive summaries). However, we changed the focus of Figure 6 and discussion to underscore the limitation of the unconventional definition of EECl that was adopted in the 2018 and 2022 Executive Summaries, which are a departure from the initial definition from Montzka et al., 1996 (Line 1191):

"Much of this work focuses on ODSs and their role in estimating EESC, which is an estimate of inorganic halogens in the stratosphere. Because stratospheric halogens originate from organic compounds that are accurately measured in the troposphere, the quantity effective equivalent chlorine (EECl) (Montzka et al. 1996) has been considered in parallel with EESC when reporting trends in ODSs. Time series of EECl have been highlighted in Figure 1 of the Executive Summary of the 2018 and 2022 SAOD reports, though an unconventional definition of EECl was used, which did not account for FRFs in the calculation. Here, we term this alternative definition equivalent tropospheric chlorine (ETCl). Figure 6a shows historical and projected values of ETCl, found using the time series for 16 principal ODSs of each SAOD report considered in this analysis (Fig. 6a). The 1980 return date of ETCl for SAODs from 2006 to 2018 are all clustered together around 2046/2047, whereas for the 2022 SAOD report the 1980 return date of ETCl is around 2054. Repeating this calculation but weighting each ODS using FRFs, following the definition of Montzka et al. (1996), we find that the EECl return dates are more evenly delayed between various SAODs, more notably in the midlatitudes than for polar regions (Figure 6b, c). The reason for this difference is that ETCl assumes FRFs of unity for all species, resulting in a larger weighting for the HCFCs than is found for either EECl or EESC. Between the 2006 and 2010 SAODs, the projected emissions of HCFC-22 dropped by a factor of ~2, reflecting the 2007 decision by the Parties to accelerate the phase-out of HCFCs (Montzka et al., 2015). For ETCl, this earlier phase-out of HCFC-22 offsets the delay in the return to 1980 value due to projected slower declines in other ODSs. While the drop in HCFC-22 makes a large contribution to ETCl, there is a substantially smaller effect on EECl (Figures 6b,c) and EESC (Figure 4) because of the use of FRFs for HCFC-22 of 0.15 and 0.44 for mid-latitude and polar air, respectively, compared to the use of FRF equal to unity for HCFC-22 and all other species in the formulation of ETCl. A proper accounting of FRFs is needed, as is done in EECl and EESC but not in ETCl, should past and projected tropospheric abundances of ODSs be used as a proxy for how future ozone depletion will be affected by anthropogenic halogens. Finally, the much longer return to 1980 dates of EESC (Figure 4) compared to EECl (Figure 6b,c) is caused by the time it takes for ODSs to reach various levels of the stratosphere as well as the distribution of these times, as is included in the age-of-air spectrum inherent in the definition of EESC."

l. 524: I do not think that they appear as an outlier. The difference has been made clear in my 2018 paper already and was thus clearly to be expected.

We've added some text to make it clear that this point was made in the reviewer's earlier paper. However, we do think our statement that the reported EESC recovery dates in the 2022 report is a notable outlier relative to the 2006 to 2018 SAODs is still true.

"Reported EESC recovery dates were relatively consistent between the 2006 and 2018 SAOD reports, with the value given in the 2022 report appearing to be an outlier, though this was in part to be expected from the adoption of a new EESC formulation (Engel et al. 2018)."

RC2:

Dear author team,

I very much enjoyed reading this well written and significant work and would recommend its publication after the following points have been addressed:

Thank you for the thorough review and many helpful suggestions!

l39-41 This is very brief introduction of a term that is central to this manuscript. There is a more comprehensive one later in the manuscript, which would be nice to link to.

We have added a reference to section 2.1 here for the reader  (Line 49).

"discussed further in Section 2.1"

l42-45 As the authors state, the MP entered into force in 1989. Why are they then only considering the last 16 years/5 assessments for their analysis?

This is a good question and worth a better explanation in the text.  There were several reasons we began with the 2006 assessment.   First, the last major update to the MP that had an appreciable effect on EESC was the 2007 Montreal amendment (including a Figure from Q14 from the 2018 SAOD's 20 questions for reference) and we were seeking to understand the reasons for the delayed EESC in the absence of major policy amendments.

[Figure]

We've added some text to explain this (Line 53):

"The reasons for this expected delay in the return of stratospheric halogens to the 1980 level have not been fully elucidated and changes to the Montreal Protocol do not explain this discrepancy as the 2007 Montreal Amendment was the last major amendment with appreciable effects on EESC (e.g., see Fig Q14-1 of Salawitch et al., 2018)."

Next, the changes in bank calculations differ substantially between 2002 and 2006. And third, we are trying to compare midlatitude and polar air between assessments, and the 2006 assessment was the first to distinguish between these in their EESC calculations (Line 102):

"We begin with the 2006 SAOD because the knowledge of the release of ODSs from banks has evolved considerably since the publication of SAOD reports prior to 2006 (IPCC/TEAP, 2005). In addition, the 2006 SAOD report is the first to distinguish the

return to 1980 for EESC of both mid-latitude and polar air, even though identical
fractional release factors for ODSs were used for both regions. For completeness, the
return to 1980 dates for EESC provided in the past eight SAOD reports are given in
Supplemental Table S1."

Apart from these reasons we chose the last five assessments to make this a tractable
paper, as we still have access to the methods from the 2006 Assessment on. We were not
confident we could explain or reproduce results from assessments prior to 2006.

l45-48 Very long sentence. Consider splitting it in two.

We have split this into two sentences.

l47 Please make it clearer to the reader, which one is the newest SAOD, and which one first
adopted the new EESC definition.

We have reworded this to be more clear (Line 56).

"However, the newer EESC formulation (Engel et al., 2018) first used in the 2018 SAOD
may play an important role, as the return to 1980 levels was delayed by more than a
decade simply from using this newer approach (WMO, 2018)."

l51 Consider changing this to "global atmospheric lifetime".

Thank you, we did this!

l73-74 This is partly an odd choice of references for proving the point of a "consistent
underestimate of the global production of ODSs". The Benish et al., paper is not global at all nor
does it estimate production or emissions, and none of the cited works focuses on the fourth
primary gas listed later in this paragraph, i.e., halon-1301.

Thank you for taking note of this. We have removed the Benish et al. citation and agree it's out
of place here. We added the Gamlen 1986 paper, to underscore the underreporting of CFC-11
over time along with the Montzka paper. The citation of the Lickley et al. 2022 paper does
provide an estimate of consistent biases in reported production of halon-1301 (along with 9 other
ODSs subject to banking).

l83-84 I encourage the author to consistently use either common names or chemical formulas. It
would also be recommendable to give both in at least one place, e.g., Table S2.

Thank you for this suggestion, and we agree this is good practice. We are now using the common
names for all gases in the main text and include their chemical formula along with their common
name in Table 2.

l84 Looking at Table 2, there are other ODSs that appear to make significant contributions (e.g.,
HCFC-22, halon-2402). The changes in these gases just changed signs during the various

assessments, resulting in a comparably small net change between the 2006 and 2022 SAOD. While I do not agree with this approach, I can certainly see how it simplifies the analysis. However, I would recommend to at least explain this simplification to the reader upfront.

Thank you for the suggestion. It is true that we had to make choices to make the analysis more tractable (as we've done on choosing the last 5 assessments instead of all assessments). We've added some text in the introduction to explain why we have not analyzed the other 12 gases with this level of detail (Line 115).

> "While the other 12 gases have led to both positive and negative changes in EESC between SAODs, their overall net contribution to the return of EESC to 1980 has been substantially smaller than the contribution of the primary four gases. Therefore, our focus is on quantifying the impact of changes in the mole fraction projections of CFC-11, CFC-12 and halon-1301 and CTC on EESC."

l112-115 How small exactly? And how does this link to the effect from "updated historical mole fraction estimates of ODSs" mentioned in the abstract?

This is a good question. In attempts to answer this question, we have compared the effect of historical mole fraction updates on EESC return dates as follows. We first calculate the return to 1980 EESC levels using the mole fractions up to 1980 from the 2006 SAOD, along with mole fractions from the SAOD 2022 for 1981 onwards. We then perform the same analysis using the entire SAOD 2022 time series. We find 2006 SAOD historical values provide a higher 1980 EESC value, and thus an earlier return date of 1.2 years than the 2022 SAOD historical values provide. We then estimated each ODSs contribution to the 1.2 year difference. We find that updated historical abundances of methyl bromide alone contributed to a 2.4 year delay in EESC return date. The next largest contributors are halon-1211 and HCFC-22 where updated historical abundances both have accelerated the return date by 1 year and 0.4 years, respectively. All other ODS historical updates result in a less than 3 month change in EESC return date (Line 176).

"However, we note here that updates of historical mole fraction abundances contributed to a 1.2 year delay in EESC return dates between the 2006 and 2022 SAOD, with historical updates to methyl bromide contributing to 2.4 years of this delay while other ODSs' historical updates (notably halon-1211 and HCFC-22) contributed to an acceleration in the return date. "

l141: "age-of-air" is not defined here nor in Table 1. There is, at this stage of the manuscript, only a brief explanation of the "age spectrum" in l134-135 and it is not made clear to the reader how this links to "age-of-air".

We have updated the text to introduce and define age-of-air when we introduce the age spectrum (Line 202):

> "$G$ represents the distribution of time required to be transported from entry into the stratosphere to the region of interest and is referred to as the age spectrum. This transport time is referred to as the age-of-air of an air parcel and represents the amount of time since the parcel was last in the troposphere (Kida 1983)."

l141-142 This is not correct - see, e.g., Box 1-4, Figure 1-18, and Table 1-7 in the 2018 SAOD. Only for projections (different chapter) was the Newman et al. method preferred.

Thank you for pointing this out.  We have edited the text to be more precise (Line 205):

"The 2014 and 2018 SAODs adopted the Newman et al. (2007) formulation of EESC projections, which modified equation (1) such that both $f_i$  and $\rho_i$ were time-weighted averages, reflecting the non-linear dependence of these terms on the age-of-air in the stratosphere.  In the 2018 SAOD, the Engel et al. (2018) formulation (which employs slightly different fractional release values and a different age spectrum), is adopted in Chapter 1 on historical estimates of EESC and in an appendix of Chapter 6, applied to future projections.  The 2022 SAOD  aopted the Engel et al. (2018) formulation for computation of EESC for both historical and future projections."

l147-148 As far as I understand the Engel et al. approach it leads to lower effective mean ages for chemical tracers as compared to inert ones.

We believe our interpretation is correct –note that Andreas Engels' was reviewer 1, and his comment on L401 above also confirms our interpretation. In the Engel et al. approach, air is weighted based on the time when the trace gas dissociates, meaning younger air, which has not yet dissociated, will have a lower weight than older air, leading to an age spectrum that is skewed towards older air relative to previous approaches.  Because this is not obvious, we've added a clarifying explanation (Line 221)

"For EESC, this change results in higher weighting of air with *longer* transit times through the stratosphere and lower weight to air with *shorter* transit times (for which ODSs have been dissociated to a lesser degree, particularly in the midlatitudes), than found using the approach of Newman et al. (2007). In summary, using the new formulation, EESC lags the troposphere more strongly than an inert tracer would."

l184-187 This is a rather confusing mixture of examples. I recommend revising the statement to better distinguish between largely satellite-based methods focusing on the determination the stratospheric burden (1st example), complete atmospheric burden and loss rate modeling approaches with input from near-ground observations (2nd example) and outdated early estimation methods of the stratospheric loss rate (3rd example). Perhaps it is also worth clarifying that the loss for CFCs and halons occurs pretty much entirely in the stratosphere.

We adopted most of these suggestions (though many of these assumptions are outdated, they were still used in to inform lifetimes in the SAODs) and agree that this paragraph needed clarification for what is currently being used (Line 376):

"These lifetime estimates were calculated using numerous lifetime inference methods (see the Stratosphere-troposphere Processes and their Role in Climate (SPARC, 2013) report for more details).  Lifetimes have been based on satellite-derived methods which convolve stratospheric distributions (as a function of altitude and pressure) of long-lived gases with photolysis rates of their destruction (Minschwaner et al., 1993), model

inversion methods using ground-based measurements with prescribed emissions (Rigby et al., 2013), or tracer-tracer methods, which relate the slope of mixing ratio of a particular species to the mixing ratio of another species with a well-established lifetime (Plumb and Ko 1992). However, for many species, modelled lifetimes alone inform the atmospheric lifetimes used in the SAODs (as reported in SPARC, 2013). For CFCs and halons, atmospheric loss occurs primarily in the stratosphere through photolysis. For gases such as methyl chloroform that undergo removal in the troposphere due to processes such as reaction with OH, the lifetime may be revised due to better knowledge of the rate constant for reaction with OH, the average OH concentration itself, as well as additional years of data from which the lifetime is inferred (Prinn et al. 2001; Montzka et al. 2011).."

l203 Coming back to my earlier point (l42-45), why was, e.g., the 2002 SAOD cited here not included in the analysis?

We hope our explanation to your earlier point has now clarified this.

l248-249 Amazing that the shaded region is so narrow. Does this create a conflict with the earlier statement of the large uncertainties in bank estimates? Also, presumably lifetime uncertainties are not included here?

It appears that the shaded region may not have shown up in the pdf version of this figure for reviewer 1, so it could be that the shaded region is missing for reviewer 2's version of the figure. We apologize. Regardless, the reviewer is right that the shaded region is likely too narrow because uncertainties in lifetimes were not accounted for the in 2022 SAOD, but rather were prescribed in order to be consistent with other chapters in the assessment.

l253 Please consider an opening statement detailing why CCl4, and CCl4 only, gets an extra section here. It is also unclear why this rather lengthy section repeats the well-known CCl4 story, which has received much attention in the past including an entire SPARC report. Is this perhaps meant as an illustrative example of how and why bank estimates and projections can change over time?

We have added some text to the beginning of this section to explain (Line 473).

"We consider CTC separately from the banked ODSs as it is not thought to be a substantially banked chemical and global emissions have been much less well understood. Therefore, CTC projections have been developed using an independent method compared to banked ODSs."

It is true that the CCl4 story has received much attention, however, our experience presenting this work to atmospheric chemists has informed us that despite this fanfare, it is still not widely known. Further, because CCl4 projections have had such a large influence on changing estimates of EESC recovery, we feel it is important to explain why.

l261 Please add "CCl4".

Done!

l267 These should probably be negative percentages. Alternatively, you could add "decreases of", although "decrease" is already used in the following sentence.

Thank you for catching this. We agree this should refer to decreases in emissions and for clarity, we think it's okay to repeat this word in the subsequent sentence as it has identical meaning.

l268 "near zero" in terms of absolute emissions?

You're right that this is a rather subjective term. We removed this and instead refer the reader to figure 2.

l277-278 Perhaps worth explaining what "feedstock" means in this context. Also, there is only one PCE.

We've added an explanation here (Line 518):

"where feedstocks refer to chemicals used in the process of manufacturing different chemicals"

l339-341 Consider shifting this sentence to the beginning of "Step 6" for narrative reasons.

Done!

L348 The title of this section is somewhat inappropriate as it contains quite a bit of discussion.

Thank you. We changed the title of this section to be Results and Discussion.

l358-361 Again, I don't think this is the correct explanation. Have a look at Figure 1-18 in the 2018 SAOD: If it were a simple shift, then the two different EESCs should not be virtually identical at polar winter conditions.

We believe this is the correct interpretation for mid-latitude EESC (which we have now specified is what we are referring to in this section). For polar EESC, however, any gas that will have dissociated, would have done so by the time it reaches the poles, so the weighting from the Engel et al. method only leads to a meaningful change in EESC in midlatitudes, where there is variability across gases with respect to the time lag in the dissociation of source gases. This is what we show in Figure 4. We have amended the text to make this point more clearly (Line 775):

"There are also large differences in the return to 1980 dates *between* the various formulations for EESC. For example, return dates of midlatitude EESC found using Engel et al. (2018) lag those of Newman et al. (2007) by 13.8 and 3.5 years for midlatitude and polar air, respectively, when using the ODS mole fraction table given in the 2022 SOAD report. The later return date of the Engel et al. (2018) formulations is largely driven by their use of a method that accounts for the relationship of tropospheric source gas trends and stratospheric chemical breakdown. Their

EESC formulation takes into account the time needed to release the halogens from their source gases. The inorganic fraction, which EESC represents, is therefore weighted towards longer transit times and thus lags the troposphere more strongly than in the older formulation. The Engel et al. (2018) approach used in the 2022 SAOD report again leads to lower EESC values during the ascending phase of the tropospheric halogen loading and higher EESC values during the descending (recovery) phase of tropospheric halogen loading, and thus to a longer time frame needed to reach 1980 EESC values."

l386 Consider giving the WMO 2022 return date in the caption as it is not included in the table.

Thank you for this suggestion, it has been added.

l404-405 It might be worth adding that the main reason behind the differences in polar and midlatitude EESC is not merely the longer residence time: For the former air predominantly arrives via the upper branch of the Brewer-Dobson circulation and has therefore been exposed to much more tropical UV photolysis, whereas the latter is influenced more by the lower branch.

Thank you for this suggestion.  We've added some clarifying text in this explanation (Line 1018):

"The formulations give similar estimates of EESC for polar regions because the transit through the stratosphere from injection (in the tropics) to polar descent includes the transit of air parcels through the upper branch of the Brewer-Dobson circulation.  This results in nearly complete loss of most ODSs, due to longer residence time in the stratosphere and most importantly exposure to a more intense ultraviolet radiation environment than is seen for most 3-year old, mid-latitude air parcels. Thus, the age of air associated with dissociated ODSs is much more similar to the age of an inert tracer in polar regions than is the case for mid-latitude air."

L485 In the version I downloaded the medium and light purple color bars appeared to be missing. Also, why is WMO 2022 not shown?

We apologize that our figure has not shown up properly.  The WMO 2022 is the end reference year in the figure, and we have added a label to the line to demarcate this year.

l491 Two consecutive sentences starting with "The next update corresponds to...".

Thank you. We removed this repetition.

l510 Other ODSs such as HCFC-141b or CH3CCl3 also have their most of their loss in the troposphere. The main reason for HCFC-22 being much less influential in the EESC is its low FRF. Consider adding this to the discussion.

We believe this is an important point raised by the reviewer that required some careful consideration on our part regarding the underlying EECl definition.  In particular, the 2018 and

2022 assessments computed EECl using FRFs of unity for all species, rather than the conventional definition from Montzka et al. 1996. We shifted the focus of Figure 6 to the comparison of EECl using the original formulation vs the unconventional definition, and clarify in the discussion that HCFC-22's FRF is an important component driving the difference between these curves. Our updated discussion of the new figure 6 is shown below (Line 1215):

"Much of this work focuses on ODSs and their role in estimating EESC, which is an estimate of inorganic halogens in the stratosphere. Because stratospheric halogens originate from organic compounds that are accurately measured in the troposphere, the quantity effective equivalent chlorine (EECl) (Montzka et al. 1996) has been considered in parallel with EESC when reporting trends in ODSs. Time series of EECl have been highlighted in Figure 1 of the Executive Summary of the 2018 and 2022 SAOD reports, though an unconventional definition of EECl was used, which did not account for FRFs in the calculation. Here, we term this alternative definition equivalent tropospheric chlorine (ETCl). Figure 6a shows historical and projected values of ETCl, found using the time series for 16 principal ODSs of each SAOD report considered in this analysis (Fig. 6a). The 1980 return date of ETCl for SAODs from 2006 to 2018 are all clustered together around 2046/2047, whereas for the 2022 SAOD report the 1980 return date of ETCl is around 2054. Repeating this calculation but weighting each ODS using FRFs, following the definition of Montzka et al. (1996), we find that the EECl return dates are more evenly delayed between various SAODs, more notably in the midlatitudes than for polar regions (Figure 6b, c). The reason for this difference is that ETCl assumes FRFs of unity for all species, resulting in a larger weighting for the HCFCs than is found for either EECl or EESC. Between the 2006 and 2010 SAODs, the projected emissions of HCFC-22 dropped by a factor of ~2, reflecting the 2007 decision by the Parties to accelerate the phase-out of HCFCs (Montzka et al., 2015). For ETCl, this earlier phase-out of HCFC-22 offsets the delay in the return to 1980 value due to projected slower declines in other ODSs. While the drop in HCFC-22 makes a large contribution to ETCl, there is a substantially smaller effect on EECl (Figures 6b,c) and EESC (Figure 4) because of the use of FRFs for HCFC-22 of 0.15 and 0.44 for mid-latitude and polar air, respectively, compared to the use of FRF equal to unity for HCFC-22 and all other species in the formulation of ETCl. A proper accounting of FRFs is needed, as is done in EECl and EESC but not in ETCl, should past and projected tropospheric abundances of ODSs be used as a proxy for how future ozone depletion will be affected by anthropogenic halogens. Finally, the much longer return to 1980 dates of EESC (Figure 4) compared to EECl (Figure 6b,c) is caused by the time it takes for ODSs to reach various levels of the stratosphere as well as the distribution of these times, as is included in the age-of-air spectrum inherent in the definition of EESC."

l511 The resolution of this figure appears to be lower than that of others in the manuscript.

Thank you for noticing this. We have adjusted the resolution in the resubmission.

l530-532 This statement is a bit misleading as the main changes to lifetime estimates did not occur in the 2022 SAOD.

We edited the statement to be more specific (Line 1603):

Lifetime assumptions in the 2006 and 2010 assessments for CFC-11 and other ODSs were lower than the current best estimates, contributing to an earlier expected return date for EESC than found using lifetimes in the 2022 SAOD report.

l537 Again (see l84 comment), it would be good to state explicitly that this is only the NET change between the 2006 and the 2022 SAOD.

Thank you, we made this change (Line 1615):

"The remaining 12 ODS mole fraction projection updates account for an additional net change of ~1 year between SAODs."

l556-559 I think this might be a bit of an overstatement. Quantitatively, how high might the "portion of the slower than expected decline of EESC" that is "caused by inadvertent atmospheric releases of CCl4 from a wide range of industrial activities" actually be?

We believe this is poorly constrained. According to the Sherry et al. 2018 paper, and reported in the 2022 SAOD Chapter 7, Figure 7-4, the unreported inadvertent emissions and unreported non-feedstock emissions for 2019 were between 5 and 20 Gg, so up to nearly half of observationally-derived emissions in 2019 come from these sources. We quantify here that 15 Gg of continued emissions beyond 2022 would contribute to a 3-year delay in EESC. So we believe the statement reflects the literature and our findings.

l577 From a brief look at Bednarz et al., I think this should be the lower rather than the upper stratosphere here.

Thank you, we made this correction.

l583 I think the EECl rather than the EESC is used to estimate "radiative forcing of climate by ODSs"?

This is a good point. We removed the reference to radiative forcing here, as it's not the focus of the present work.

l589 It's not necessarily the EESC return dates that are changing. To a large degree it is merely the estimates/projections of these.

This is true. We've amended the text accordingly.

l594-596 Given that this very work demonstrates that a substantial part of the accumulated delay stems from an improved formulation of EESC, it might be worth tempering this statement a bit. In addition, it might be worth at least mentioning the potential impact of anthropogenically driven global warming, which is expected to lead to changes to, e.g., stratospheric circulation therefore inducing further "atmospheric uncertainties".

We understand that our work does show that scientific uncertainties explain some of these delays – though some of the atmospheric updates, such as lifetimes updates, may be biased due to the assumed emissions used to derive them.  So, we feel this statement is appropriate for the context of this work, however, we modified the sentence by implying that atmospheric uncertainties should only be a part of the explanation of consistent delays (Line 1703):

> "However, we would expect that each new update in the representation of atmospheric processes could lead to either accelerations or delays in expected EESC return times. Changes in expected EESC return dates resulting mainly in delays are not expected to be likely a result of atmospheric uncertainties alone, but rather may suggest the potential for either breaches in the Protocol or significant emissions resulting from the use of ODSs as feedstock, which are not controlled by the Protocol."

We agree that anthropogenic influences on circulation are relevant to future updates.  We had attempted to acknowledge this referring to the Brewer-Dobson circulation, though have edited the text to make it clear this with regards to global warming (Line 1669):

> "Future updates may include changes in estimated ODS atmospheric residence times resulting, for example, from anthropogenic global warming driven changes to the Brewer-Dobson circulation (Prather et al., 2023; Fleming et al. 2011), ocean exchanges, and changes in the hydroxyl radical (Wang et al. 2023)."

l596-598 A very nice closing statement. Given that it is not certain that the EESC delay trend is going to continue in the future, the use of the subjunctive might be more appropriate, though.

Thank you for this suggestion, we have changed this to the subjunctive.